# Rethinking Temporal Consistency in Video Object-Centric Learning: From Prediction to Correspondence

**Zhiyuan Li** [1]   **Rongzhen Zhao** [1]   **Wenyan Yang** [1]   **Wenshuai Zhao** [2]   **Pekka Marttinen** [2]   **Joni Pajarinen** [1]

## Abstract

The de facto approach in video object-centric learning maintains temporal consistency through learned dynamics modules that predict future object representations, called slots. We demonstrate that these predictors function as expensive approximations of discrete correspondence problems. Modern self-supervised vision backbones already encode instance-discriminative features that distinguish objects reliably. Exploiting these features eliminates the need for learned temporal prediction. We introduce Grounded Correspondence, a framework that replaces learned transition functions with deterministic bipartite matching. Slots initialize from salient regions in frozen backbone features. Frame-to-frame identity is maintained through Hungarian matching on slot representations. The approach requires zero learnable parameters for temporal modeling yet achieves competitive performance on MOVi-D, MOVi-E, and YouTube-VIS. Project page: https://magenta-sherbet-85b101.netlify.app/

## 1. Introduction

Unsupervised object-centric learning decomposes visual scenes into discrete slot representations (Locatello et al., 2020). The field has evolved from synthetic benchmarks toward unconstrained real-world video sequences, where maintaining object identity across temporal sequences has become a central problem. The dominant solution treats this challenge through predictive dynamics modeling: methods deploy transitioners that forecast future slot states from historical context (Kipf et al., 2022; Elsayed et al., 2022; Zadaianchuk et al., 2023; Aydemir et al., 2023).

Current implementations share two design choices. Slot initialization remains content-blind: methods either sample queries from a shared Gaussian distribution or learn fixed parameters applied uniformly across datasets (Locatello et al., 2020; Kipf et al., 2022; Manasyan et al., 2025; Qian et al., 2023). Temporal consistency then relies on learned dynamics modules, ranging from recurrent networks to temporal Transformers and physics-informed architectures (Kossen et al., 2020; Singh et al., 2022; Wu et al., 2023a; Meo et al., 2024; Li et al., 2025b; Wu et al., 2023b; Li et al., 2025a). While these methods achieve competitive benchmark performance, they assume that maintaining object identity requires explicit simulation of physical dynamics. This assumption carries substantial computational cost.

We challenge the necessity of learned dynamics modules. These high-capacity predictors often reduce to identity permutations, functioning as computationally expensive solutions to what is fundamentally an alignment problem. Modern self-supervised vision backbones already encode object-binding signals within their feature maps (Li et al., 2025c). Patches belonging to the same instance exhibit elevated feature similarity, effectively creating unsupervised saliency maps where high-confidence regions correspond to object centers. When the encoder already provides instance-aware features, temporal consistency becomes a problem of maintaining stable slot-to-object correspondences rather than simulating forward dynamics.

Temporal consistency is a correspondence problem, not a prediction problem. Slot Attention exhibits permutation equivariance (Locatello et al., 2020), meaning that semantic content is independent of index order. Maintaining object identity across frames requires only that we recover the correct permutation between consecutive timesteps. A parameter-free bipartite matching solver suffices for this task. Combined with initialization grounded in backbone saliency, this eliminates the need for learned dynamics modules.

We introduce Grounded Correspondence, a framework that replaces parameterized temporal transitioners with determin-

[1]Department of Electrical Engineering and Automation, Aalto University, Finland [2]Department of Computer Science, Aalto University, Finland. Correspondence to: Zhiyuan Li <zhiyuan.li@aalto.fi>.

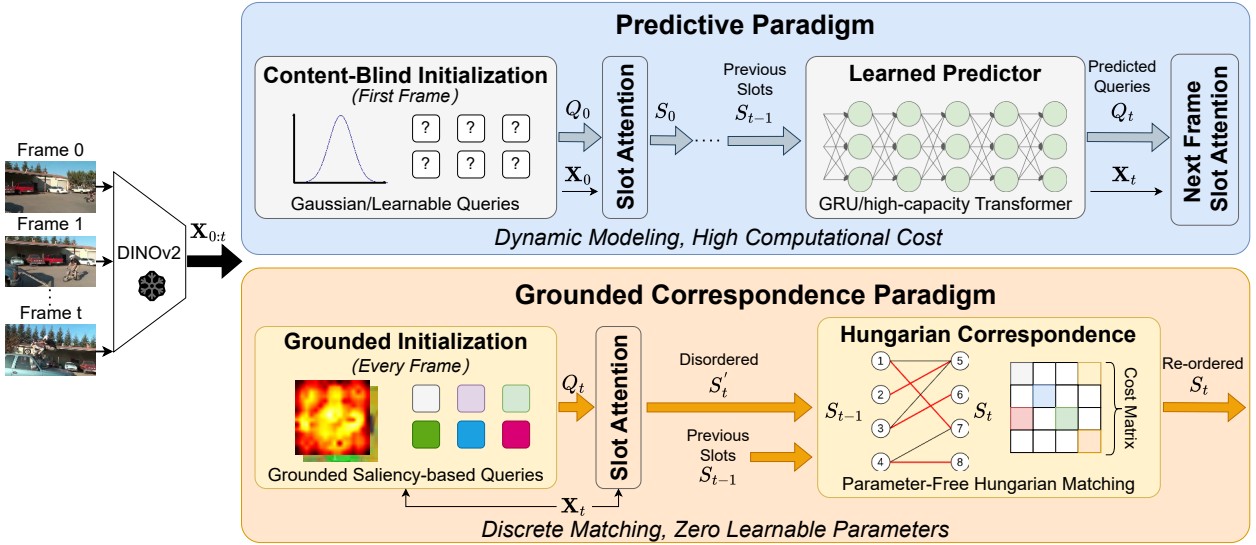

*Figure 1.* **From prediction to correspondence: rethinking temporal consistency in video object-centric learning. Top:** The predictive paradigm uses content-blind initialization without spatial priors. Temporal consistency relies on learned dynamics modules that forecast future slot states through high-capacity Transformers (Vaswani et al., 2017) or recurrent networks (Cho et al., 2014). **Bottom:** Grounded Correspondence eliminates learned temporal parameters entirely. Slots initialize from saliency peaks in frozen backbone features. Frame-to-frame identity is maintained through parameter-free Hungarian matching (Kuhn, 1955) on slot representations. This paradigm shift achieves competitive performance with zero learnable parameters for temporal modeling.

istic discrete optimization. Slot queries are initialized from emergent saliency peaks in the frozen backbone. Frame-to-frame identity is maintained through Hungarian matching (Kuhn, 1955) on slot features. The approach achieves competitive performance on MOVi-D, MOVi-E, and YouTube-VIS while requiring zero learnable parameters for temporal modeling. We review related work in Section 2, analyze current methods to motivate our design in Sections 3 to 4, present the framework in Section 5, and provide experimental validation in Section 6 (preliminaries in Appendix A). Our contributions are as follows:

- We show that exploiting instance-aware signals from self-supervised backbones DINOv2 (Oquab et al., 2024) accelerates slot convergence compared to content-blind initialization, reducing the computational cost of the iterative grouping process.

- We demonstrate empirically that learned temporal transitioners in current methods approximate solutions to bipartite matching problems, suggesting architectural redundancy.

- We introduce Grounded Correspondence, a framework combining grounded initialization with parameter-free Hungarian matching that requires zero learnable parameters for temporal modeling.

- Through extensive experiments on MOVi-D, MOVi-E, and YouTube-VIS, we demonstrate strong improve-

ments on synthetic benchmarks (+15.7 ARI on MOVi-D, +6.8 ARI on MOVi-E) and competitive performance on real-world sequences.

## 2. Related Work

**Video Object-Centric Learning.** Unsupervised scene decomposition extracts slot representations from visual data. Early methods reconstructed pixels on synthetic sequences (Kipf et al., 2022), while subsequent work targeted feature-level reconstruction on naturalistic videos (Singh et al., 2022; Elsayed et al., 2022; Zadaianchuk et al., 2023). Applications now include datasets with unconstrained visual variability (Aydemir et al., 2023) and downstream tasks including 3D novel-view synthesis (Liu et al., 2025b), language-controllable editing (Didolkar et al., 2025), and multiple-object tracking (Zhao et al., 2023). Existing methods enforce object-centric structure through specialized architectural components. We examine whether modern vision backbones already encode sufficient structural priors to reduce this architectural complexity.

**Slot Initialization Strategy.** Slot Attention requires initializing query vectors that determine the resulting decomposition quality. Standard methods sample from a shared Gaussian distribution (Kipf et al., 2022) or use fixed learnable parameters (Manasyan et al., 2025). MetaSlot (Liu et al., 2025a) introduces a prototype codebook to handle variable slot counts, while SMTC (Qian et al., 2023) em-

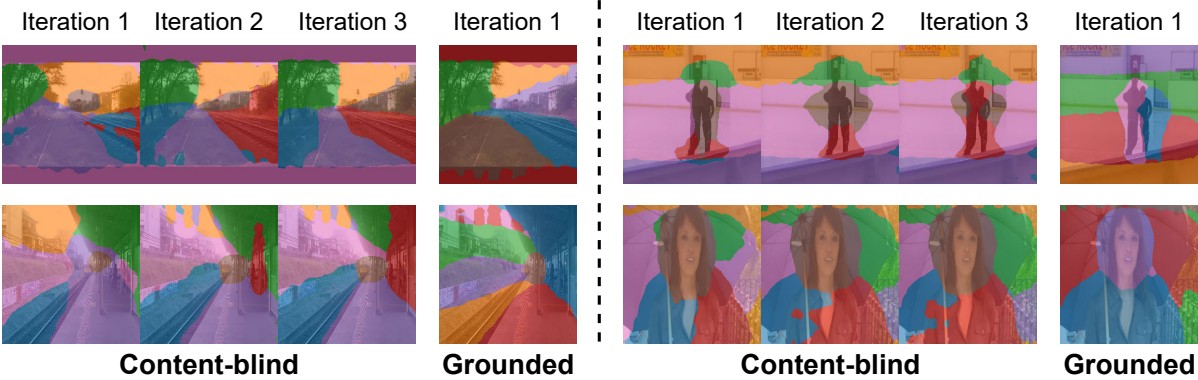

*Figure 2.* **Slot Attention convergence with different initialization strategies.** Four examples from YouTube-VIS. For each example, left shows content-blind initialization across 3 iterations, right shows grounded initialization with 1 iteration. Content-blind initialization requires multiple iterations to stabilize object boundaries, while grounded initialization achieves stable segmentation in a single iteration.

ploys semantic-aware Gaussian means. These approaches remain agnostic to image content at initialization. Recent work demonstrates that self-supervised Vision Transformers encode object-binding signals within their feature maps: patches from the same instance exhibit elevated similarity (Li et al., 2025c). SlotContrast (Manasyan et al., 2025) briefly explores k-means clustering for initialization. We exploit these emergent instance-aware properties directly, seeding slots at high-saliency feature locations rather than learning content-agnostic priors.

**Temporal Consistency.** Existing methods treat temporal consistency as a prediction task. Models deploy recurrent transitioners (Kossen et al., 2020; Lin et al., 2020) or temporal Transformers (Wu et al., 2023a; Li et al., 2025a; Singh et al., 2022; Kipf et al., 2022) to forecast future slot states. Variants incorporate conditional autoregressive priors (Meo et al., 2024), physics-informed Hamiltonian constraints (Li et al., 2025b), or latent diffusion for slot-conditioned denoising (Wu et al., 2023b). Recent work refines query prediction mechanisms (Zhao et al., 2025) or distills representations into efficient student architectures (Maus & Maki, 2025; Grigore et al., 2025). These approaches assume that maintaining identity requires modeling object dynamics. We argue that temporal consistency is a correspondence problem rather than a prediction problem. When the visual encoder provides instance-discriminative features, discrete matching between consecutive frames suffices to maintain object identity. We replace learned temporal predictors with deterministic bipartite matching.

## 3. Rethinking Slot Initialization: From Content-Blind to Grounded

Slot Attention partitions visual tokens through iterative competitive grouping (Locatello et al., 2020). In video object-

centric learning, slots for frame $t$ are refined through multiple attention iterations conditioned on the state from frame $t - 1$ (see Appendix A for detailed formulation). Current methods typically apply three iterations for the initial frame and two iterations for subsequent frames to ensure stable decompositions. This iterative overhead stems from architectural choices rather than fundamental requirements of the binding mechanism. When slot initialization exploits the structural priors already present in vision backbone features, the grouping process converges substantially faster.

### 3.1. The Iterative Search Bottleneck

Standard video object-centric learning methods apply Slot Attention with three iterations for the initial frame and two for subsequent frames. SAVi (Kipf et al., 2022) and SlotContrast (Manasyan et al., 2025) follow this pattern to ensure mask stability. This computational overhead is often justified as necessary for unsupervised grouping. The requirement stems from content-blind initialization. Methods that initialize queries from fixed learnable vectors (Liu et al., 2025a) or random Gaussian distributions (Locatello et al., 2020) must use early iterations to locate objects in feature space. Without spatial or semantic priors, the model searches for salient regions before grouping can begin.

We evaluate initialization strategies using SlotContrast (Manasyan et al., 2025) under controlled conditions. Content-blind initialization uses three iterations for the initial frame and two for subsequent frames, following standard practice. Grounded initialization requires only one iteration for both initial and subsequent frames. Figure 2 shows that fixed learnable slots require multiple iterations to resolve object boundaries, while grounded initialization achieves stable segmentation faster. Additional qualitative examples are provided in Appendix C.1. Table 1 confirms that grounded initialization with reduced iterations outper-

*Table 1.* Performance comparison of content-blind and grounded initialization on YouTube-VIS using SlotContrast. Content-blind uses 3 iterations for the initial frame and 2 for subsequent frames. Grounded uses 1 iteration for all frames. Mean and standard deviation over 3 seeds. Grounded initialization improves segmentation quality across all metrics.

| Initialization | ARI ↑ | FG-ARI ↑ | mBO ↑ |
|---|---|---|---|
| Content-blind | $32.1_{0.8}$ | $36.3_{0.2}$ | $29.9_{0.3}$ |
| Grounded | $\mathbf{34.6_{0.7}}$ | $\mathbf{37.5_{0.5}}$ | $\mathbf{31.2_{0.5}}$ |

*Table 2.* Performance of independent per-frame discovery versus SlotContrast on YouTube-VIS. Mean and standard deviation over 3 seeds. Image-level metrics show comparable segmentation quality; video-level metrics reveal inconsistent identity tracking.

| Evaluation | Method | ARI ↑ | mBO ↑ |
|---|---|---|---|
| Image-level | Independent Discovery | $41.7_{1.7}$ | $34.8_{0.2}$ |
| | SlotContrast | $41.9_{0.7}$ | $38.1_{0.5}$ |
| Video-level | Independent Discovery | $12.1_{2.1}$ | $17.1_{1.3}$ |
| | SlotContrast | $32.1_{0.8}$ | $29.9_{0.3}$ |

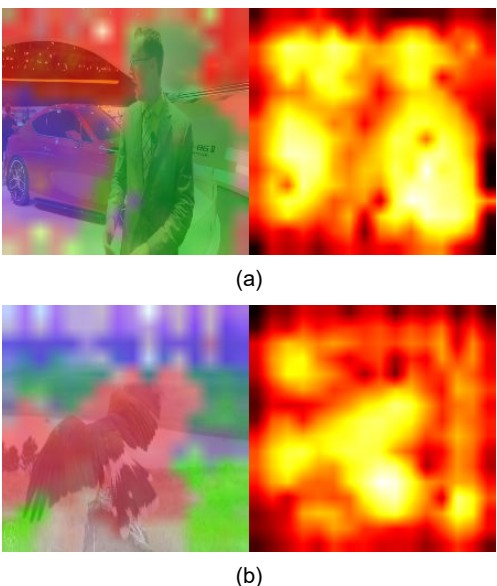

(a)

(b)

*Figure 3.* **Emergent object-binding signals in DINOv2 features.** Left: Principal component analysis (PCA) visualization of patch embeddings shows coherent spatial clustering by object instance. Right: Saliency map derived from feature similarities (brighter regions indicate higher saliency). Peaks correspond to object centers.

forms content-blind baselines quantitatively. These results indicate that much of the iterative overhead stems from poor initialization rather than fundamental requirements of the binding process. Proper initialization provides slots with sufficient spatial and semantic information to accelerate object grouping.

### 3.2. Emergent Objectness and Saliency-Based Initialization

Self-supervised Vision Transformers encode object-binding signals within their feature maps (Li et al., 2025c). Patch embeddings from the same object instance exhibit elevated similarity, a property that emerges from the pretraining objective without explicit supervision. These similarity patterns create implicit saliency maps where high-confidence

regions correspond to object centers. Figure 3 visualizes this emergent objectness: peaks in the feature similarity map align with object centroids. Additional examples demonstrating this property across diverse scenes are provided in Appendix C.2. Initializing slot queries at these salient locations, described in Section 5, accelerates the convergence of Slot Attention. Current architectures use iterative grouping to discover object structure that the visual encoder has already encoded in its features.

### 3.3. The Discrepancy Between Per-Frame Discovery and Temporal Stability

We compare an independent discovery baseline against Slot-Contrast (Manasyan et al., 2025). The baseline decomposes each frame independently without temporal recurrence. Table 2 shows that on image-level segmentation metrics, independent discovery matches the performance of temporal models. The visual encoder provides sufficient information to resolve object boundaries within individual frames without temporal context.

However, video-level tracking metrics reveal a critical limitation. Figure 4 illustrates the failure mode: while the model segments objects correctly in each frame, slot assignments permute randomly across time. The segmentation quality remains high, but object identities are not preserved. Additional examples of this index permutation phenomenon are provided in Appendix C.3. Frame-level discovery succeeds through emergent backbone features, but temporal consistency requires an additional mechanism. The question becomes whether this mechanism must be a learned dynamics predictor or whether simpler index alignment suffices.

## 4. Rethinking Temporal Consistency: Prediction or Correspondence?

Existing methods assume that maintaining object identity across frames requires learned predictors that model dynamics. In this section, we test whether temporal consistency can instead be solved through correspondence: matching slot representations between consecutive frames without

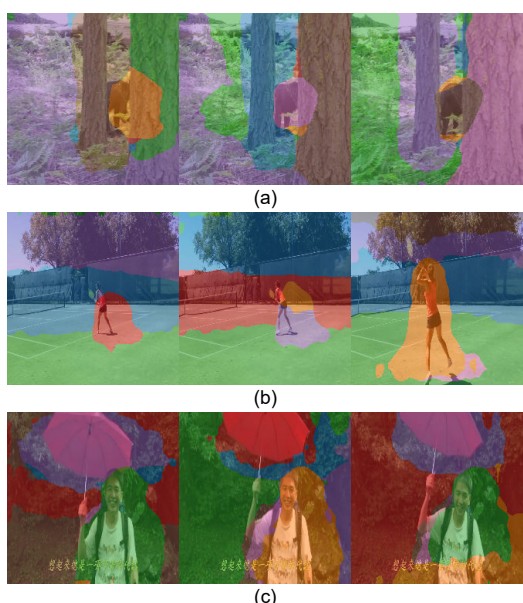

(a)

(b)

(c)

*Figure 4.* **Index permutation in independent discovery.** Three YouTube-VIS video sequences. Each row shows consecutive frames from one sequence. Objects are segmented correctly in each frame, but slot assignments (indicated by colors) change randomly across time, demonstrating inconsistent identity tracking.

*Table 3.* SlotContrast with and without learned temporal prediction on YouTube-VIS. The identity baseline $Q_t = S_{t-1}$ matches the performance of the full model. Mean and standard deviation over 3 seeds.

| Temporal Module | ARI ↑ | FG-ARI ↑ | mBO ↑ |
|---|---|---|---|
| Learned predictor | $32.1_{0.8}$ | $36.3_{0.2}$ | $29.9_{0.3}$ |
| Identity ($Q_t = S_{t-1}$) | $\mathbf{33.0_{1.1}}$ | $\mathbf{36.6_{0.7}}$ | $\mathbf{30.5_{0.6}}$ |

predicting future states.

## 4.1. The Empirical Redundancy of Temporal Predictors

We evaluate SlotContrast (Manasyan et al., 2025) against a variant where the learned transitioner $\phi_\theta$ is removed. In this ablation, queries for frame tt t are simply the slots from frame $t - 1$: $Q_t = S_{t-1}$. This identity mapping replaces the learned predictor entirely. Table 3 shows that removing the predictor causes minimal performance degradation on YouTube-VIS; this finding generalises across pretrained backbones (DINOv1, DINOv2, DINOv3), as detailed in Appendix B and Table 8. Learned transitioners provide little benefit over the identity baseline. This result aligns with recent findings on temporal architectures (Zhao et al., 2025) and suggests that current methods dedicate substantial computational resources to a component that contributes marginally to performance.

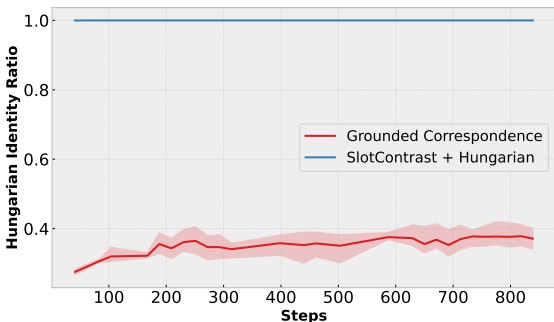

*Figure 5.* Hungarian Identity Ratio on YouTube-VIS equals 1.0 when slots are propagated as identity between frames, indicating that discrete matching suffices for temporal consistency.

## 4.2. The Permutation Hypothesis: Learning Index Stability

Slot Attention is permutation equivariant: reordering the input queries produces correspondingly reordered outputs. Temporal consistency, therefore, requires preventing slots from permuting randomly between frames. We argue that learned predictors accomplish this by approximating an identity function. Rather than modeling object dynamics or forecasting future states, these modules primarily stabilize slot indices against permutation symmetry. Section 3 established that vision backbones already provide instance-discriminative features. When features distinguish objects reliably, using large Transformer architectures to approximate identity mappings is computationally wasteful.

## 4.3. Verification of Temporal Correspondence via Discrete Solvers

We test the Permutation Hypothesis by replacing learned predictors with the Hungarian algorithm described in Appendix A.3. We define the Hungarian Identity Ratio: the fraction of slots where optimal bipartite matching between frames $t$ and $t - 1$ yields an identity assignment, $i = \sigma(i)$. We compute this ratio by solving the Linear Sum Assignment Problem on slot feature similarities.

Figure 5 compares two scenarios on YouTube-VIS sequences. When slots from frame $t - 1$ are propagated as queries for frame $t$, as in the SlotContrast variant with Hungarian matcher, the identity ratio equals 1.0. Slots remain sufficiently distinct in feature space that simple propagation maintains consistent assignments without learned prediction. In contrast, our method reinitializes slots independently for each frame from grounded saliency, causing the ratio to drop below 1.0. This demonstrates that slot ordering varies across frames. The Hungarian matcher successfully realigns these permuted indices. This comparison confirms two insights: learned predictors in existing methods approximate identity mappings when slots propagate naturally, while discrete

correspondence matching handles the reordering introduced by independent initialization.

# 5. Method: Grounded Correspondence Framework

We introduce Grounded Correspondence, a framework that separates object discovery from temporal tracking. Existing methods use learned transition functions $\phi_\theta$ and initialize slots without considering image content. Grounded Correspondence instead exploits instance-aware features from a frozen Vision Transformer backbone. The framework has two components. Grounded Saliency Initialization identifies object locations by finding peaks in the backbone feature similarity map. Hungarian Correspondence maintains identity across frames through bipartite matching on slot features.

## 5.1. Grounded Discovery via Emergent Feature Consistency

The foundational premise of our discovery mechanism is that self-supervised vision backbones establish a latent topology where feature proximity encodes instance identity. Following Li et al. (2025c), we characterize the feature map $\mathbf{F} \in \mathbb{R}^{N \times D}$ as a collection of $N$ patch embeddings $\{f_i\}_{i=1}^N$. We define the Emergent Binding Hypothesis: the cosine similarity $\mathbf{A}_{i,j} = \frac{f_i \cdot f_j}{\|f_i\|\|f_j\|}$ represents a non-parametric kernel density estimator (Elgammal et al., 2002) for the shared identity of patches $i$ and $j$. To identify robust object centroids without supervision, we formulate the discovery task as finding the local modes of the binding distribution.

**Local Instance Consistency** ($L_i$)**.** We model the objectness of a patch $i$ as its local evidence density within the feature manifold. Under the assumption that objects are semantically and spatially contiguous, the local consistency $L_i$ is defined as the mean kernel density over a spatial neighborhood $\mathcal{N}_i$ of radius $r$:

$$L_i = \frac{1}{|\mathcal{N}_i|} \sum_{j \in \mathcal{N}_i} \mathbf{A}_{i,j}.$$

In the context of manifold learning, $L_i$ identifies the medoids of local clusters. Patches with high $L_i$ reside at the centers of high-density feature regions, while patches at boundaries exhibit low density due to high feature variance.

**Global Redundancy Suppression** ($G_i$)**.** To ensure that discovered centroids represent distinct entities rather than pervasive background structures, we introduce a global contrast term. We define the global background prior as the centroid of the entire feature distribution $\bar{f} = \frac{1}{N} \sum_{k=1}^N f_k$. The global similarity $G_i$ represents the projection of patch $i$ onto this common background mode:

$$G_i = \frac{f_i \cdot \bar{f}}{\|f_i\|\|\bar{f}\|}.$$

In the latent space of a ViT, $G_i$ typically captures background elements (e.g., sky or low-frequency textures) that dominate the global mean. Penalizing $G_i$ acts as a whitening-like transformation (Koivunen & Kostinski, 1999), emphasizing patches that are discriminative relative to the scene average.

**Grounded Saliency Metric** ($S_i$)**.** Combining these density-based estimators, we define the Grounded Saliency Metric as:

$$S_i = L_i - \alpha \cdot G_i,$$

where $\alpha$ modulates the background penalty. By identifying the local maxima of the field $\mathbf{S} \in \mathbb{R}^N$, we deterministically retrieve the existing backbone knowledge to seed the initial query matrix $\mathbf{Q}_t$, replacing random or learned initialization with direct extraction from backbone features.

## 5.2. Grounded Saliency Initialization

To extract a set of $K$ queries that are both individually significant and collectively diverse, we implement a greedy optimization process within the feature manifold. This mechanism ensures that slots are initialized at distinct object centroids and prevents the grouping process from collapsing onto redundant semantic entities.

**Iterative Mode Selection.** We iteratively identify the patch index $p$ that attains the global maximum of the current saliency field:

$$p = \arg \max_{i \in \{1...N\}} S_i.$$

The feature vector $f_p$ becomes the query seed for the $k$-th slot. Each slot thus initializes at a salient location in the feature map.

**Feature Suppression.** To enforce diversity across the query set, we update the saliency map immediately following each selection. We utilize the cosine similarity between the selected centroid $f_p$ and the remaining feature set $\mathbf{F}$ to suppress redundant saliency values:

$$S \leftarrow S \cdot (1 - \text{clamp}(\text{sim}(f_p, \mathbf{F}), 0, 1)).$$

This suppression removes the selected object region from the saliency map. Iterating this process prevents selecting multiple queries from the same object. The resulting query matrix $\mathbf{Q}_t$ distributes slots across distinct object instances.

## 5.3. Hungarian Correspondence

Section 4 established that learned temporal predictors are unnecessary. We replace them with a bipartite matching.

When slots are propagated unchanged from frame $t-1$ to frame $t$, identity is preserved automatically. However, grounded initialization recomputes slot locations each frame, potentially discovering objects in different orders. Matching resolves this by aligning slot indices to maintain consistent object identities.

**Cost Matrix.** For consecutive frames at timesteps $t-1$ and $t$, we obtain slot representations $\mathbf{s}_{t-1}$ and $\mathbf{s}_t$. We construct a cost matrix $\mathbf{C} \in \mathbb{R}^{K \times K}$ where each element $C_{i,j}$ denotes the cosine distance between slots:

$$C_{i,j} = 1 - \frac{\mathbf{s}_{t-1,i} \cdot \mathbf{s}_{t,j}}{\|\mathbf{s}_{t-1,i}\|\|\mathbf{s}_{t,j}\|}$$

This metric captures the semantic displacement of object representations in the latent manifold between frames.

**Hungarian Alignment.** To establish a globally optimal temporal assignment, we solve the LSAP as defined in Appendix A.3. We find the optimal permutation $\pi^*$ that minimizes the total alignment cost across the slot set:

$$\pi^* = \arg \min_{\pi \in \mathcal{P}_K} \sum_{i=1}^{K} C_{i,\pi(i)}$$

The resulting permutation $\pi^*$ re-orders the current slots $\mathbf{S}_t$ to align with the previous frame's slot indices. This discrete matching approach maintains temporal consistency without learned dynamics modules.

# 6. Experiments

We evaluate Grounded Correspondence on synthetic and real-world video benchmarks. Our experiments address three questions: How does the framework perform without learned temporal parameters compared to state-of-the-art methods? Which initialization strategy provides the most stable object discovery? How do the components of the Grounded Saliency Metric, specifically local density estimation, background suppression penalty, and spatial aggregation radius, affect performance across different datasets?

## 6.1. Experimental Setup

**Benchmarks.** We evaluate our framework on MOVi-D and MOVi-E from the Kubric synthetic dataset (Greff et al., 2022), and on the real-world YouTube-VIS 2021 dataset (Yang et al., 2021). MOVi-D features high object density with both static and dynamic objects. MOVi-E introduces linear camera movement. YouTube-VIS 2021 consists of unconstrained video sequences.

**Metrics.** Following standard protocol in video object-centric learning, we employ the Adjusted Rand Index (ARI) and Foreground ARI (FG-ARI) to evaluate segmentation quality. We also report Mean Best Overlap (mBO) to measure mask decomposition accuracy.

**Implementation Details.** Our framework uses a frozen DINOv2 backbone to extract instance-aware features. The Grounded Saliency Initialization and Hungarian Correspondence components require no learnable parameters for temporal modeling. All hyperparameter settings are provided in Appendix D.

## 6.2. Does Grounded Correspondence Achieve Competitive Performance Without Learned Temporal Parameters?

Table 4 compares Grounded Correspondence against Slot-Contrast (Manasyan et al., 2025) across synthetic and real-world benchmarks. Our framework requires zero learnable parameters for temporal modeling yet demonstrates strong performance on synthetic data and competitive results on real-world sequences. Qualitative results are provided in Appendix C.4.

On synthetic benchmarks, Grounded Correspondence significantly outperforms the baseline: +15.7 ARI on MOVi-D and +6.8 ARI on MOVi-E. These results validate our hypothesis from Section 4 that learned temporal predictors become unnecessary when object discovery leverages emergent backbone features. However, Grounded Correspondence produces slightly less sharp masks on synthetic data, reflected by lower mBO scores. On YouTube-VIS 2021, Grounded Correspondence achieves competitive performance, approaching baseline results on FG-ARI and mBO. The results demonstrate that discrete correspondence matching suffices for temporal consistency without parametric dynamics modeling, even on unconstrained sequences.

## 6.3. Which Initialization Strategy Provides the Most Stable Object Discovery?

We compare the Grounded Saliency Metric against two alternative initialization strategies: Norm-based saliency, which uses the norm of feature embeddings, and PCA-based saliency, which ranks patches by their projection magnitude onto the top eigenvectors of the feature covariance matrix.

Table 5 compares three initialization strategies. On MOVi-E, PCA-based initialization improves over Norm-based (+2.4 FG-ARI), while the Grounded Saliency Metric achieves larger gains over PCA-based (+3.8 FG-ARI) and Norm-based (+6.2 FG-ARI). On YouTube-VIS, the Grounded Saliency Metric outperforms PCA-based (+0.4 FG-ARI) and Norm-based (+1.6 FG-ARI). These results confirm that density-based mode finding provides more reliable object localization than global statistical measures.

*Table 4.* Performance on synthetic MOVi datasets and real-world YouTube-VIS. Grounded Correspondence uses zero learnable parameters for temporal modeling yet demonstrates strong improvements on synthetic benchmarks and competitive results on unconstrained sequences.

| | MOVi-D | | | MOVi-E | | | YouTube-VIS | | |
|---|---|---|---|---|---|---|---|---|---|
| | ARI ↑ | FG-ARI ↑ | mBO ↑ | ARI ↑ | FG-ARI ↑ | mBO ↑ | ARI ↑ | FG-ARI ↑ | mBO ↑ |
| SlotContrast (Manasyan et al., 2025) | $58.0_{0.8}$ | $58.0_{0.8}$ | $\mathbf{30.0_{0.4}}$ | $68.9_{0.9}$ | $68.9_{0.9}$ | $\mathbf{26.6_{0.6}}$ | $\mathbf{32.1_{0.8}}$ | $\mathbf{36.3_{0.2}}$ | $\mathbf{29.9_{0.3}}$ |
| GROUNDED CORRESPONDENCE | $\mathbf{73.7_{1.7}}$ | $\mathbf{73.7_{1.7}}$ | $28.4_{0.3}$ | $\mathbf{75.7_{1.4}}$ | $\mathbf{75.7_{1.4}}$ | $23.4_{0.5}$ | $30.1_{4.4}$ | $33.1_{1.6}$ | $29.3_{1.9}$ |

*Table 5.* Initialization strategies for slot discovery on MOVi-E and YouTube-VIS. The Grounded Saliency Metric outperforms norm-based and PCA-based alternatives, providing more stable object localization than global feature statistics.

| | MOVi-E | | YouTube-VIS | |
|---|---|---|---|---|
| | FG-ARI ↑ | mBO ↑ | FG-ARI ↑ | mBO ↑ |
| Norm-based | $69.5_{10.3}$ | $21.6_{3.3}$ | $31.5_{1.1}$ | $28.4_{0.9}$ |
| PCA-based | $71.9_{5.1}$ | $22.7_{1.2}$ | $32.7_{0.8}$ | $27.6_{0.8}$ |
| Ours | $\mathbf{75.7_{1.4}}$ | $\mathbf{23.4_{0.5}}$ | $\mathbf{33.1_{1.6}}$ | $\mathbf{29.3_{1.9}}$ |

*Table 6.* Effect of background penalty $\alpha$ on segmentation quality. Optimal values are dataset-dependent: $\alpha = 1.0$ for MOVi-E and $\alpha = 0.5$ for YouTube-VIS. Excessive penalization at $\alpha = 1.5$ causes performance collapse on both benchmarks.

| | MOVi-E | | YouTube-VIS | |
|---|---|---|---|---|
| $\alpha$ | FG-ARI ↑ | mBO ↑ | FG-ARI ↑ | mBO ↑ |
| 0.5 | $73.0_{5.4}$ | $21.7_{1.3}$ | $\mathbf{33.1_{1.6}}$ | $\mathbf{29.3_{1.9}}$ |
| 1.0 | $\mathbf{75.7_{1.4}}$ | $\mathbf{23.4_{0.5}}$ | $\mathbf{33.1_{0.8}}$ | $27.4_{1.3}$ |
| 1.5 | $59.7_{6.5}$ | $20.9_{1.5}$ | $26.8_{0.3}$ | $17.2_{2.1}$ |

*Table 7.* Spatial neighborhood radius $r$ for local consistency computation. Optimal values are dataset-dependent: $r = 1$ for MOVi-E and $r = 2$ for YouTube-VIS. Large radii cause instability on synthetic data by violating spatial contiguity assumptions.

| | MOVi-E | | YouTube-VIS | |
|---|---|---|---|---|
| $r$ | FG-ARI ↑ | mBO ↑ | FG-ARI ↑ | mBO ↑ |
| 1 | $\mathbf{75.7_{1.4}}$ | $\mathbf{23.4_{0.5}}$ | $33.1_{1.0}$ | $28.6_{1.2}$ |
| 2 | $72.7_{2.6}$ | $22.0_{1.8}$ | $\mathbf{33.1_{1.6}}$ | $\mathbf{29.3_{1.9}}$ |
| 3 | $49.5_{18.6}$ | $18.2_{5.5}$ | $33.6_{1.3}$ | $29.0_{2.0}$ |

### 6.4. How Do the Grounded Saliency Components Affect Performance Across Datasets?

We examine the sensitivity of object discovery to the background suppression penalty $\alpha$ and the spatial aggregation radius $r$. These parameters control the balance between local density estimation and global redundancy suppression in the Grounded Saliency Metric formulated in Section 5.1.

**Background Suppression Penalty.** We compare three $\alpha$ values across both benchmarks (Table 6). On MOVi-E, increasing from $\alpha = 0.5$ to $\alpha = 1.0$ yields improvements (+2.7 FG-ARI and +1.7 mBO), while $\alpha = 1.5$ causes catastrophic collapse. On YouTube-VIS, both $\alpha = 0.5$ and $\alpha = 1.0$ achieve similar performance, indicating reduced sensitivity on real-world data. Optimal background penalization is dataset-dependent, with synthetic scenes tolerating stronger suppression than unconstrained sequences.

**Spatial Aggregation Radius.** We compare three $r$ values across both benchmarks (Table 7). On MOVi-E, $r = 1$ achieves 75.7 FG-ARI. Increasing to $r = 2$ yields degradation (-3.0 FG-ARI and -1.4 mBO), while $r = 3$ causes severe collapse (-26.2 FG-ARI and -5.2 mBO). On YouTube-

VIS, both $r = 1$ and $r = 2$ achieve similar performance, with $r = 3$ maintaining competitive results. MOVi-E benefits from tighter spatial neighborhoods due to well-defined object boundaries, while YouTube-VIS tolerates broader aggregation for variable object scales.

## 7. Conclusion

We have examined the role of learned temporal predictors in video object-centric learning. Our analysis demonstrates that these modules function primarily as expensive solutions to a discrete correspondence problem rather than as models of physical dynamics. When slot initialization exploits instance-discriminative features from pretrained backbones, temporal consistency reduces to bipartite matching between consecutive frames. Grounded Correspondence achieves competitive performance using discrete optimization in place of learned prediction modules. This work identifies architectural redundancies in current methods and demonstrates that temporal consistency can be maintained without parametric dynamics modeling.

**Limitations.** The framework has three main limitations. First, it does not handle occlusions: the Hungarian solver maintains correspondence between visible objects but cannot recover identities after reappearance. Second, the fixed slot count limits applicability to scenes with variable or high object density. Third, the Hungarian algorithm has $O(K^3)$ complexity in the number of slots $K$. This overhead is negligible when $K$ is small, but could become problematic for applications requiring hundreds of slots. Future work should address these constraints.

## Acknowledgements

This work was supported by the Research Council of Finland, Flagship program Finnish Center for Artificial Intelligence (FCAI), and the Research Council of Finland (357301, 358246). We acknowledge CSC – IT Center for Science, Finland, for awarding this project access to the LUMI supercomputer, owned by the EuroHPC Joint Undertaking, hosted by CSC (Finland) and the LUMI consortium through CSC. We acknowledge the computational resources provided by the Aalto Science-IT project.

## Impact Statement

This paper presents work whose goal is to advance the field of video object-centric learning. There are many potential societal consequences of our work, none of which we feel must be specifically highlighted here.

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

# A. Preliminaries

We briefly review Slot Attention, the predictive paradigm for temporal consistency, and the Linear Sum Assignment Problem.

## A.1. Slot Attention and Permutation Equivariance

Following the formulation in Locatello et al. (2020) and Liu et al. (2025a), Slot Attention (SA) maps a set of $N$ input features $\mathbf{X} \in \mathbb{R}^{N \times D}$ to a set of $K$ slots $\mathbf{S} \in \mathbb{R}^{K \times D}$. The mechanism utilizes a set of query vectors $\mathbf{Q}$ (initially sampled from a distribution or learned) to compete for input tokens via an iterative softmax-based attention mechanism over the slot dimension. For each iteration, the attention weights $\mathbf{A}$ are computed as:

$$\mathbf{A}_{i,j} = \frac{\exp(M_{i,j})}{\sum_{l=1}^{K} \exp(M_{l,j})}, \quad \text{where } \mathbf{M} = \frac{1}{\sqrt{D}} \mathbf{Q}\mathbf{K}^\top \tag{1}$$

where $\mathbf{K}$ and $\mathbf{V}$ are linear projections of the input features $\mathbf{X}$. The slots are updated via a Gated Recurrent Unit (GRU) (Cho et al., 2014) using the weighted mean of the values $\mathbf{V}$.

A fundamental property of this mechanism is permutation equivariance with respect to the queries. Let $\pi$ denote a permutation operator acting on the set of indices $\{1, \ldots, K\}$. The Slot Attention operator satisfies:

$$SA(\pi(\mathbf{Q}), \mathbf{X}) = \pi(SA(\mathbf{Q}, \mathbf{X})) \tag{2}$$

This property implies that the semantic content of the extracted slots is independent of their index order. However, in video sequences, this equivariance necessitates a mechanism to "anchor" specific object identities to consistent slot indices across time, or the model will suffer from stochastic index drifting.

## A.2. Predictive Paradigm

Current video object-centric learning methods maintain temporal consistency by generating queries from previous slot states (Kipf et al., 2022; Wu et al., 2023a). In this framework, the queries $\mathbf{Q}_t$ for the current frame are not sampled independently; instead, they are generated as a function of the slots from the previous timestep:

$$\mathbf{Q}_t = \phi_\theta(\mathbf{S}_{t-1}) \tag{3}$$

where $\phi_\theta$ is a learnable transitioner, such as a GRU (Kipf et al., 2022) or a high-capacity Transformer (Wu et al., 2023a). The objective of $\phi_\theta$ is to predict the future state of the objects to provide a warm-start for the attention mechanism. Our work challenges the necessity of this learnable $\phi_\theta$, posing it as an expensive approximation of a discrete matching problem.

## A.3. The Linear Sum Assignment Problem

The challenge of establishing a consistent one-to-one mapping between two sets of vectors can be formally characterized as the Linear Sum Assignment Problem (LSAP). Given two sets of $K$ elements, $\mathcal{U} = \{\mathbf{u}_1, \ldots, \mathbf{u}_K\}$ and $\mathcal{V} = \{\mathbf{v}_1, \ldots, \mathbf{v}_K\}$, we define a cost matrix $\mathbf{C} \in \mathbb{R}^{K \times K}$. Each entry $\mathbf{C}_{i,j} = d(\mathbf{u}_i, \mathbf{v}_j)$ represents the pairwise dissimilarity between elements under a distance metric $d(\cdot, \cdot)$, such as the negative cosine similarity.

The LSAP seeks an optimal permutation $\sigma \in \mathcal{P}_K$ that minimizes the total assignment cost:

$$\sigma^* = \arg \min_{\sigma \in \mathcal{P}_K} \sum_{i=1}^{K} \mathbf{C}_{i,\sigma(i)} \tag{4}$$

This combinatorial optimization problem is optimally solved using the Hungarian algorithm (Kuhn, 1955). In our work, we demonstrate that this discrete matching framework provides a parameter-free alternative to the learnable transition functions $\phi_\theta$ that represent the current standard in video OCL (Kipf et al., 2022; Wu et al., 2023a).

# B. Predictor Ablation Across Pretrained Backbones

To verify that the predictor-removal finding of Table 3 is not specific to a single self-supervised feature extractor, we repeat the SlotContrast YouTube-VIS predictor ablation across three pretrained Vision Transformer backbones of identical capacity:

*Table 8.* SlotContrast with and without learned temporal prediction on YouTube-VIS, across DINOv1/v2/v3 backbones. The identity baseline $Q_t = S_{t-1}$ matches the full model across all three pretrained backbones. Mean and standard deviation over 3 seeds.

| Backbone | Temporal Module | ARI | FG-ARI | mBO |
|---|---|---|---|---|
| DINOv1 | Identity ($Q_t = S_{t-1}$) | 28.45 ± 1.02 | 34.21 ± 0.51 | 25.55 ± 0.75 |
| | Transformer encoder | 28.48 ± 1.88 | 34.34 ± 0.75 | 24.62 ± 0.72 |
| DINOv2 | Identity ($Q_t = S_{t-1}$) | 33.69 ± 1.15 | 37.61 ± 0.87 | 30.67 ± 0.35 |
| | Transformer encoder | 33.15 ± 1.15 | 36.74 ± 0.25 | 29.78 ± 0.33 |
| DINOv3 | Identity ($Q_t = S_{t-1}$) | 18.94 ± 0.64 | 52.23 ± 2.11 | 6.48 ± 0.08 |
| | Transformer encoder | 25.98 ± 5.74 | 40.14 ± 12.28 | 21.75 ± 13.29 |

DINOv1, DINOv2, and DINOv3 (all ViT-B, 768-dim features). For each backbone we train SlotContrast for 100k steps with and without the learned transformer-encoder predictor (the identity variant sets $Q_t = S_{t-1}$), repeating each configuration for three seeds $\{0, 1, 2\}$. The input resolution and patch size match each backbone's native pretraining configuration: DINOv1 and DINOv3 at $512 \times 512$ with patch 16, and DINOv2 at $518 \times 518$ with patch 14. All other hyperparameters are held fixed across backbones. Results are summarised in Table 8.

For DINOv1 and DINOv2 the identity baseline is statistically indistinguishable from the learned predictor across all three metrics: the per-seed standard deviations exceed the identity-vs-learned gap on every entry, mirroring our main-text finding. For DINOv3 the learned predictor exhibits substantially elevated inter-seed variance ($\pm 5.74$ ARI, $\pm 12.28$ FG-ARI, $\pm 13.29$ mBO), while the identity baseline remains tightly clustered; the two variants overlap within two standard deviations on every metric. Together, these results indicate that the learned predictor does not provide a robust performance benefit over plain state propagation, irrespective of the chosen pretrained backbone.

**Comparison with the main-text DINOv2 entry.** There is a small numerical discrepancy between the DINOv2 row of Table 8 and the corresponding DINOv2 row of Table 3 in the main paper. The cross-backbone runs reported here use a fresh seed cohort ($\{0, 1, 2\}$) and were retrained end-to-end under a single unified pipeline shared across all three backbones, whereas the main-text Table 3 results were produced with an earlier code revision and a different seed selection. The qualitative conclusion (identity $\approx$ learned) is unchanged.

## C. Additional Qualitative Results

### C.1. Initialization Convergence

Figures 6 and 7 show additional examples of Slot Attention convergence comparing content-blind and grounded initialization strategies on YouTube-VIS. Each example demonstrates that grounded initialization achieves stable object segmentation in a single iteration, while content-blind initialization requires multiple iterations to converge.

### C.2. Emergent Saliency Visualization

Figure 8 provides additional examples of emergent object-binding signals in DINOv2 features across diverse YouTube-VIS scenes. For each example, the left image shows the PCA visualization of patch embeddings, where spatially coherent color regions indicate feature clustering by object instance. The right image shows the corresponding saliency map computed from local feature similarities. Across varied visual contexts, including indoor scenes, outdoor environments, different lighting conditions, and varying object scales, the saliency maps consistently produce high-confidence peaks at object centers. This demonstrates that the emergent objectness property generalizes broadly and provides reliable initialization signals for slot-based object discovery.

### C.3. Index Permutation in Independent Discovery

Figure 9 provides additional examples demonstrating the index permutation problem in independent per-frame discovery. Each row shows consecutive frames from a YouTube-VIS video sequence. While the spatial segmentation quality remains consistently high across frames, the slot assignments indicated by colors change randomly between consecutive timesteps. For instance, an object assigned to the orange slot in one frame may be reassigned to the green or blue slot in the next

frame, even though its visual appearance and spatial location remain stable. This stochastic permutation occurs because independent discovery performs slot attention separately for each frame without any temporal anchoring mechanism. The examples span diverse scenarios, including different object types, motion patterns, and scene complexities, demonstrating that index permutation is a fundamental limitation of frame-independent approaches rather than an artifact of specific visual conditions.

### C.4. Qualitative Comparison with SlotContrast

Figures 10, 11, and 12 show qualitative comparisons between Grounded Correspondence and SlotContrast on MOVi-D, MOVi-E, and YouTube-VIS respectively. Each figure displays consecutive video frames with segmentation masks overlaid, where different colors indicate different slot assignments.

On synthetic benchmarks (MOVi-D and MOVi-E), Grounded Correspondence produces compact object masks that assign single objects to single slots. In contrast, SlotContrast exhibits a consistent tendency to fragment scenes into over-segmented parts, splitting both foreground objects and background regions into multiple disconnected components. This over-segmentation behavior manifests across diverse synthetic scenes. The compact representation achieved by our method directly contributes to the substantial ARI improvements over the baseline on these benchmarks, as ARI penalizes incorrect partitioning of unified objects.

On real-world YouTube-VIS sequences, both methods achieve competitive performance despite the increased visual complexity. Grounded Correspondence maintains stable identity tracking across frames without learned temporal predictors, demonstrating that discrete correspondence matching suffices for temporal consistency on unconstrained data.

## D. Hyperparameter Settings

Table 9 lists the hyperparameters used for Grounded Correspondence across all three benchmarks. For the SlotContrast baseline, we use the hyperparameters reported in the original paper (Manasyan et al., 2025), with the only modification being batch size reduced from 64 to 8 due to computational constraints. All other training settings, including learning rate, optimizer, and architectural choices, remain identical to the published configuration.

Iteration 1  Iteration 2  Iteration 3   Iteration 1     Iteration 1  Iteration 2  Iteration 3   Iteration 1

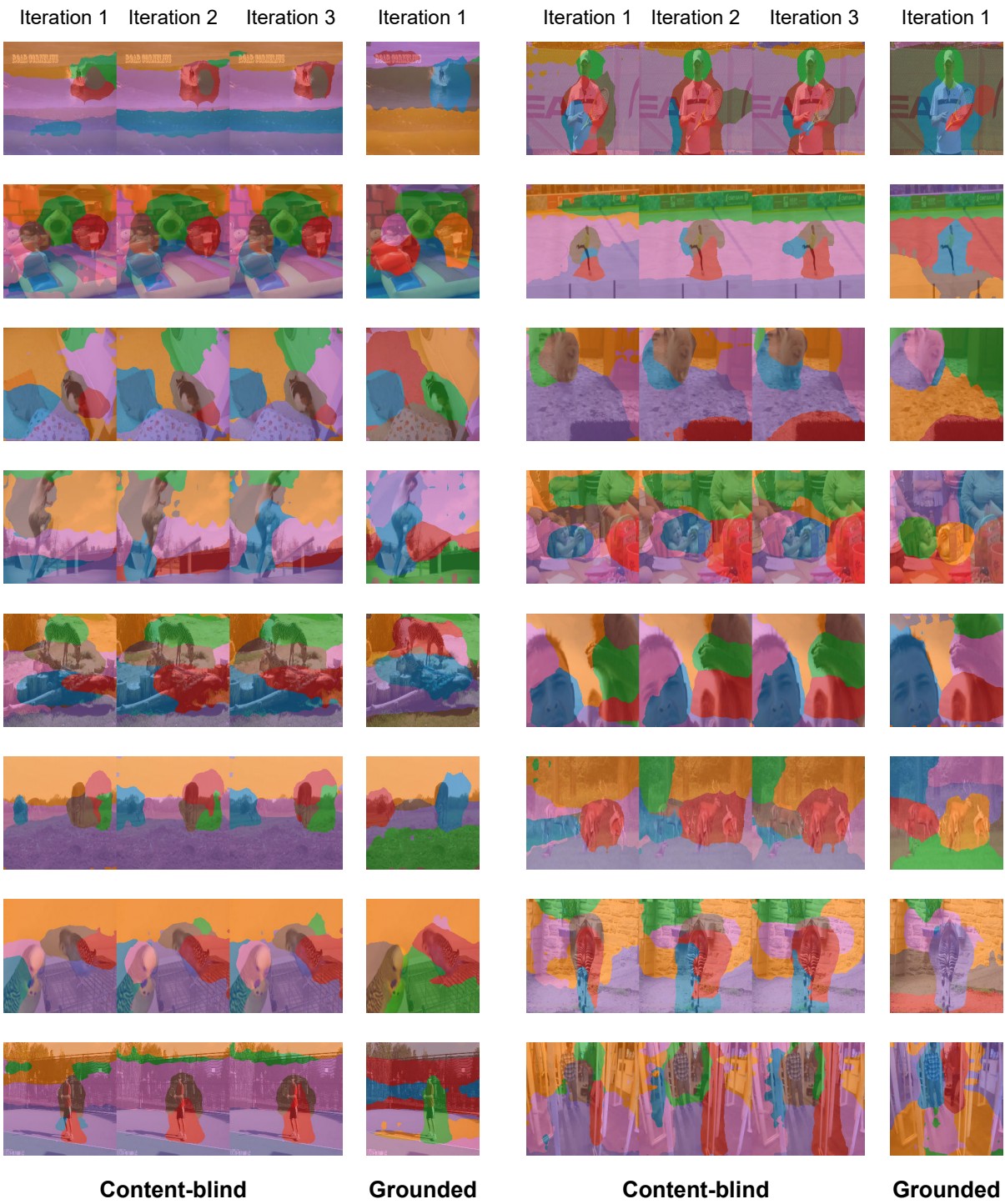

**Content-blind**     **Grounded**      **Content-blind**     **Grounded**

*Figure 6.* **Additional Slot Attention convergence examples (Part 1).** Each row: one YouTube-VIS scene. Left: content-blind initialization (3 iterations). Right: grounded initialization (1 iteration).

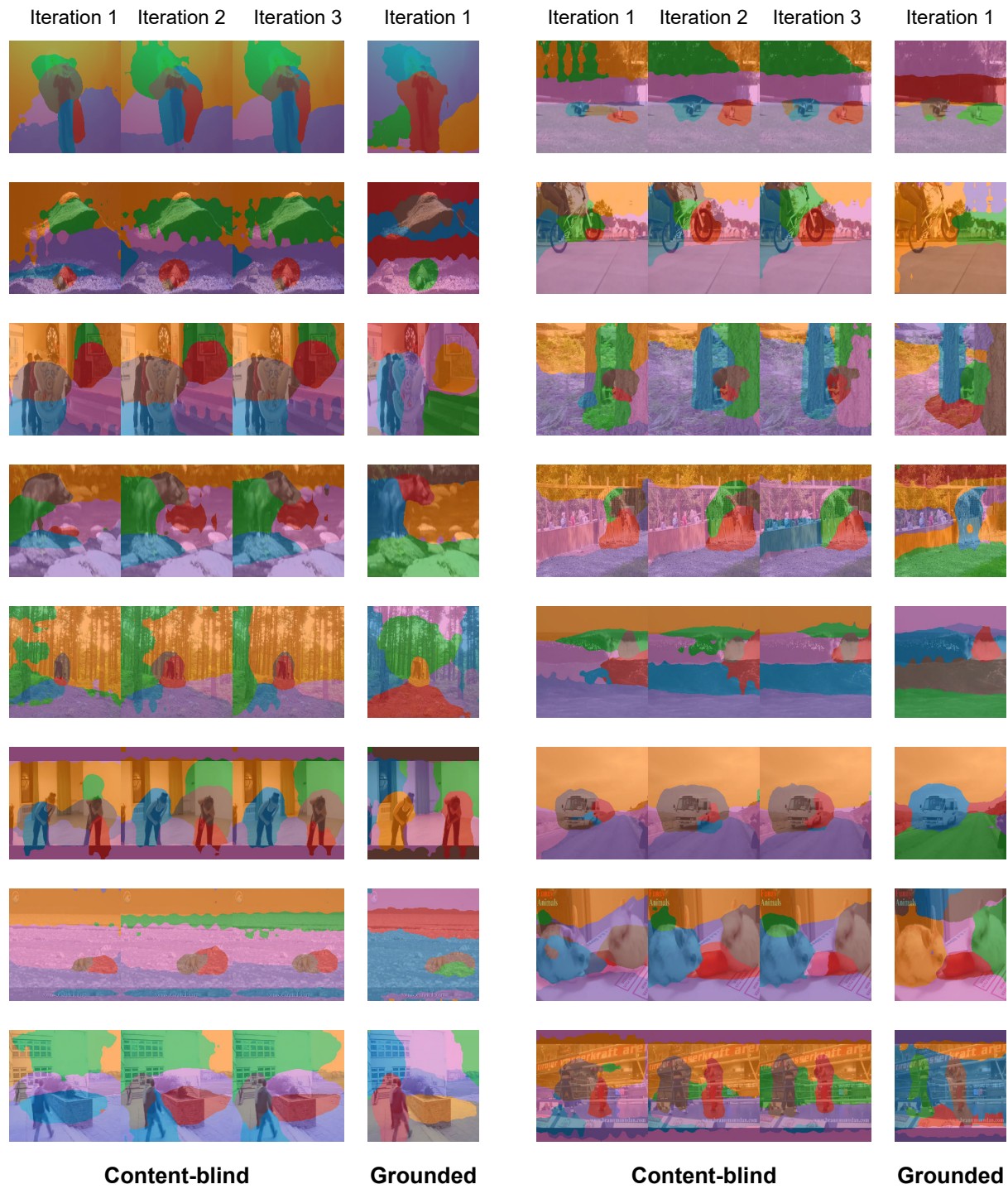

*Figure 7.* **Additional Slot Attention convergence examples (Part 2).** Each row: one YouTube-VIS scene. Left: content-blind initialization (3 iterations). Right: grounded initialization (1 iteration).

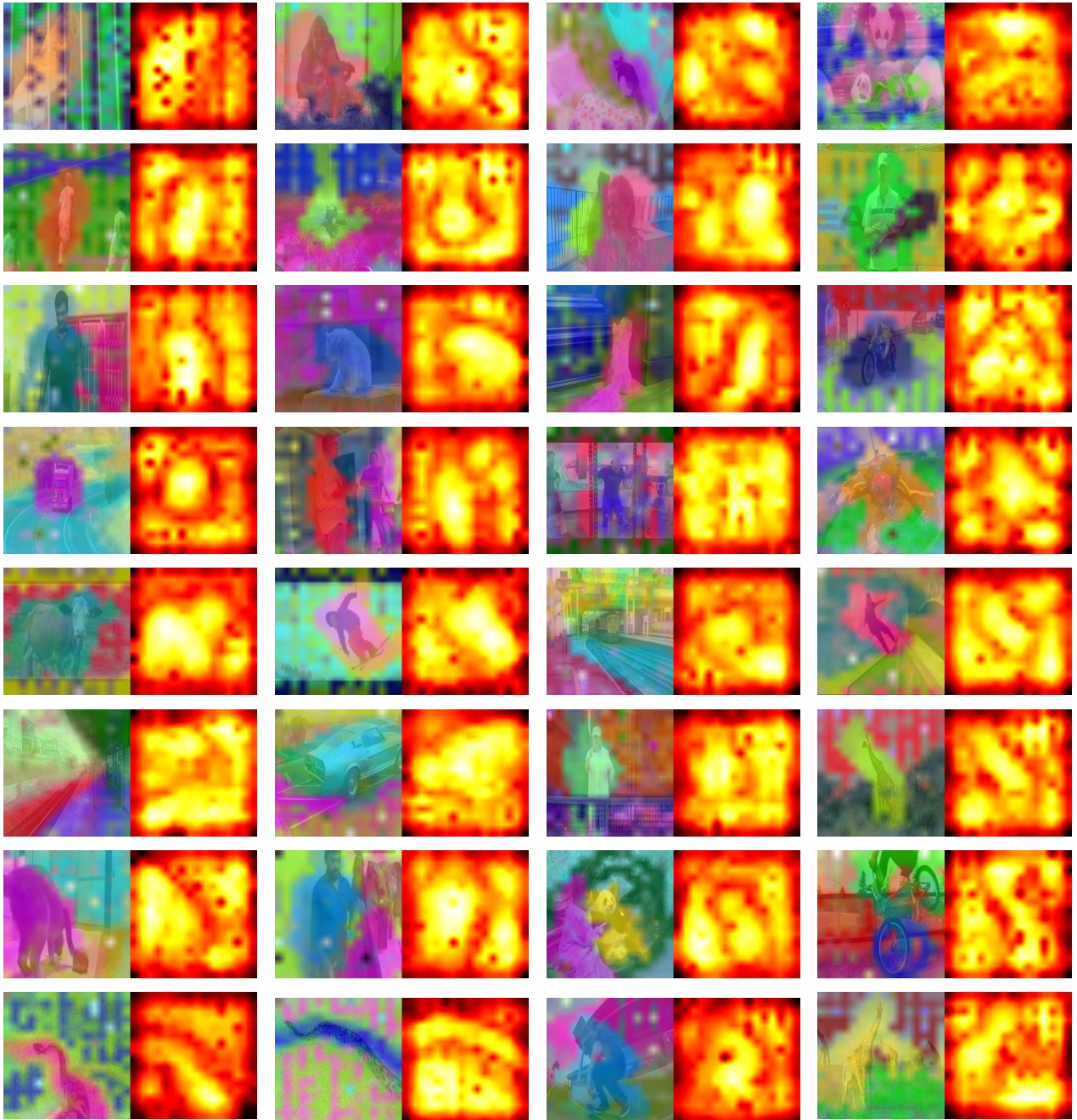

*Figure 8.* **Additional examples of emergent object-binding signals.** Each pair shows: Left: PCA visualization of DINOv2 patch embeddings. Right: Saliency map (brighter indicates higher saliency). Peaks consistently align with object centers across diverse scenes.

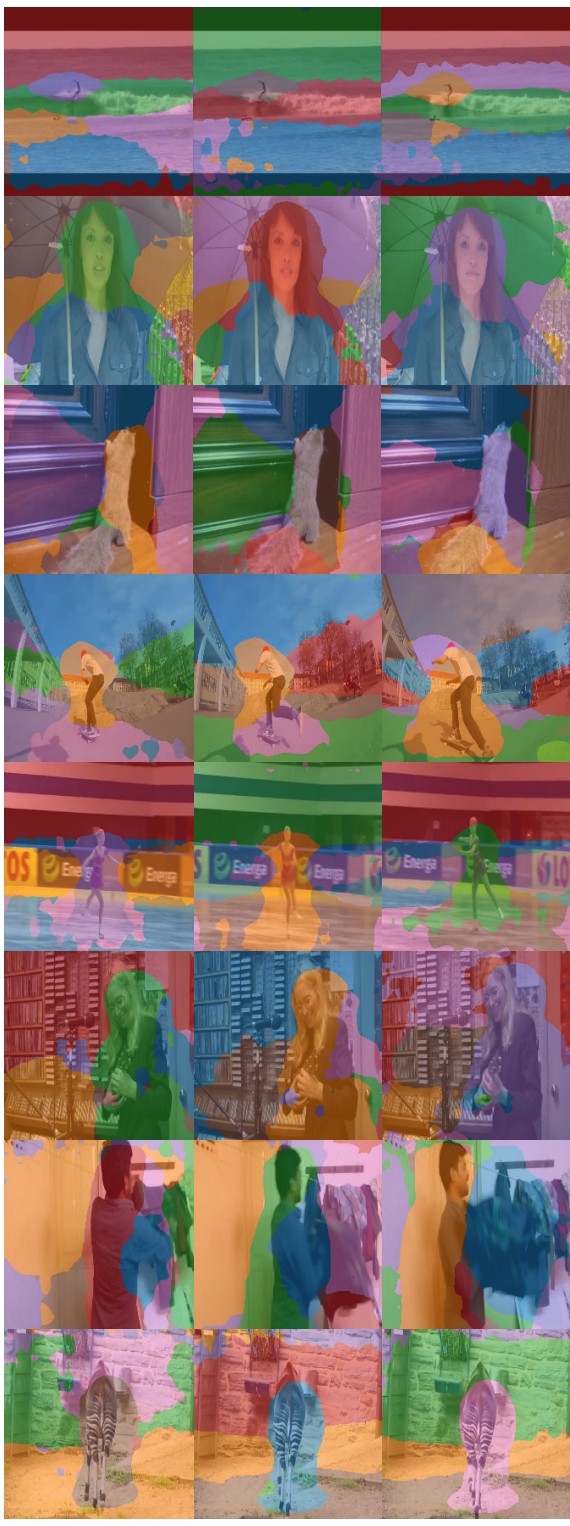

*Figure 9.* **Additional examples of index permutation.** Each row: three consecutive frames from one YouTube-VIS sequence. Objects maintain consistent segmentation quality, but slot assignments (colors) permute randomly across frames.

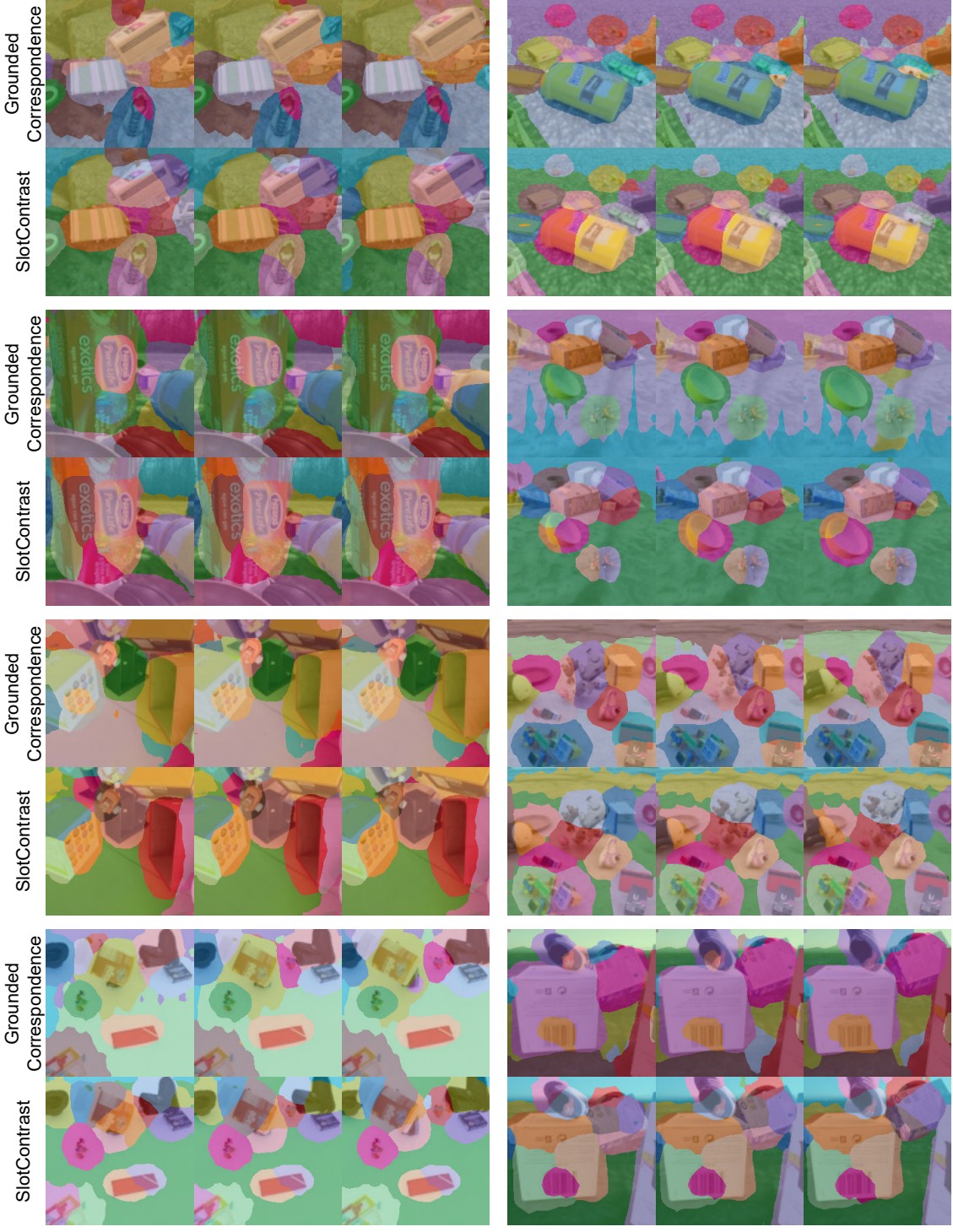

*Figure 10.* **Qualitative comparison on MOVi-D.** Each row shows consecutive frames from one sequence. Top: Grounded Correspondence. Bottom: SlotContrast. Different colors indicate different slot assignments. Grounded Correspondence produces more compact object masks, assigning single objects to single slots. In contrast, SlotContrast exhibits a tendency to split individual objects into multiple parts, as seen in several sequences where unified objects in our method correspond to fragmented, multi-colored regions in the baseline. This over-segmentation behavior aligns with the lower ARI scores achieved by SlotContrast on this benchmark.

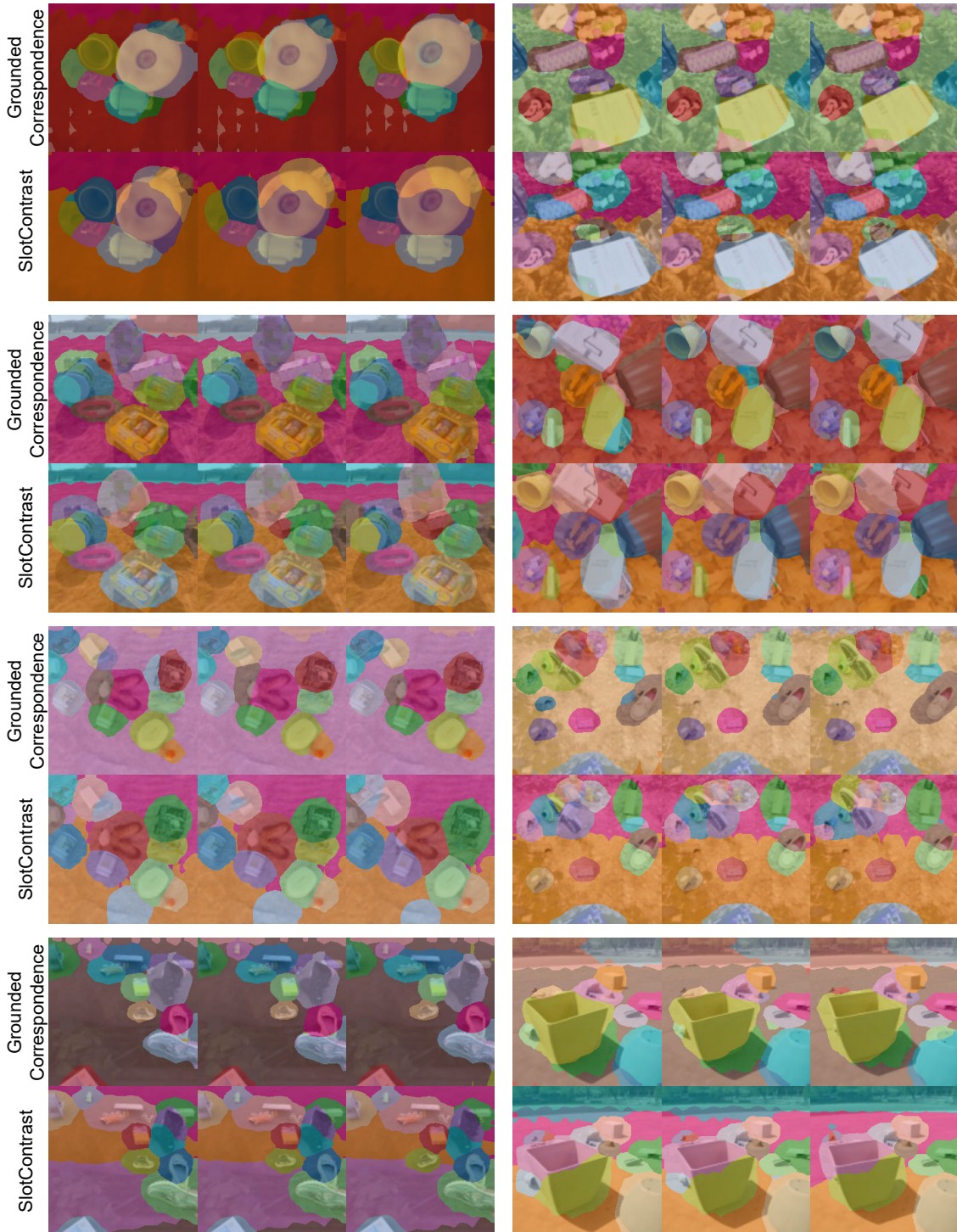

*Figure 11.* **Qualitative comparison on MOVi-E.** Each row shows consecutive frames from one sequence. Top: Grounded Correspondence. Bottom: SlotContrast. Different colors indicate different slot assignments. Grounded Correspondence generates compact masks that unify objects and background regions into coherent segments. SlotContrast exhibits fragmentation across both foreground and background, splitting continuous surfaces into multiple disconnected parts. The compact representation achieved by our method contributes to the substantial ARI improvement over the baseline on this benchmark.

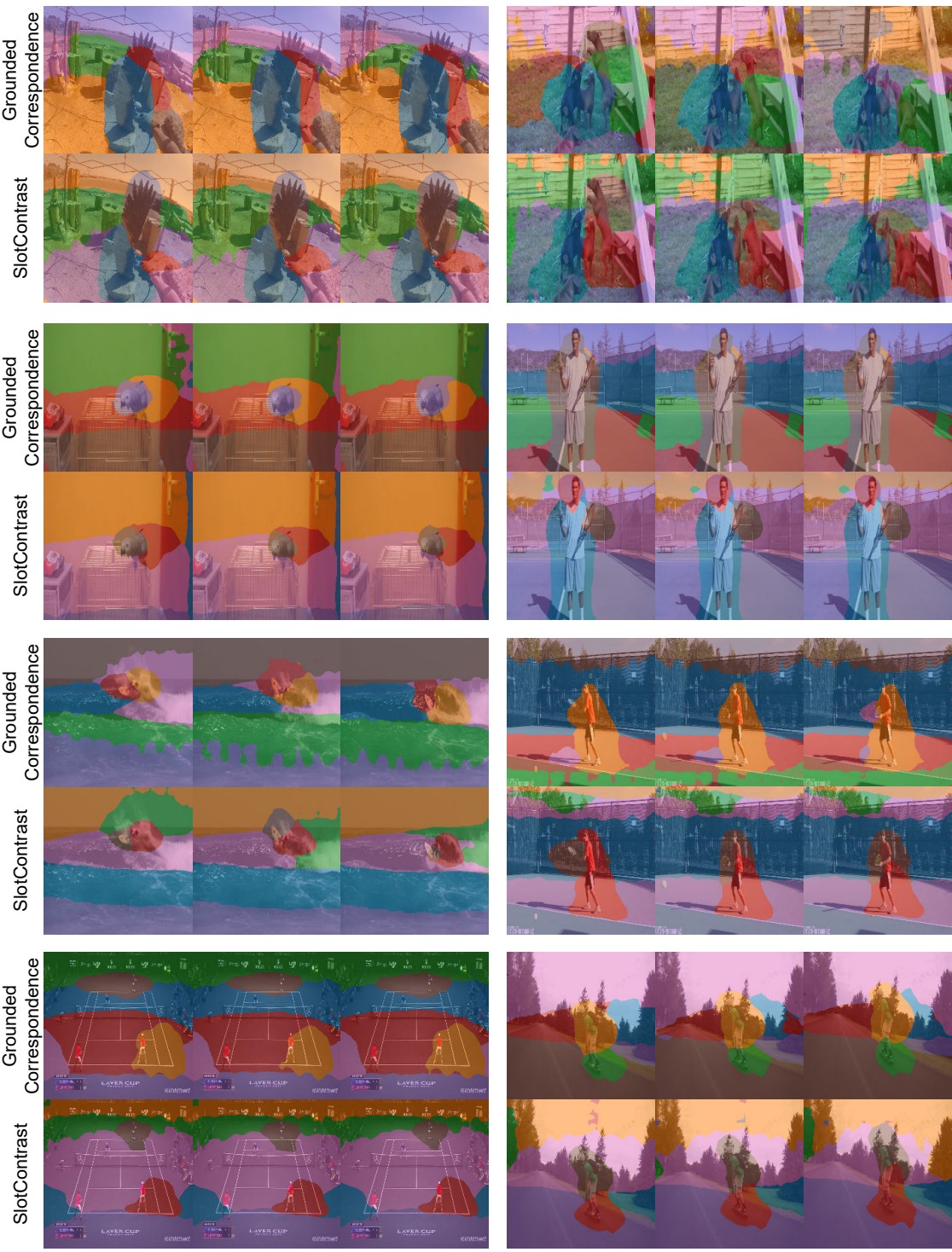

*Figure 12.* **Qualitative comparison on YouTube-VIS.** Each row shows consecutive frames from one sequence. Top: Grounded Correspondence. Bottom: SlotContrast. Both methods achieve competitive performance on unconstrained real-world sequences.

*Table 9.* Hyperparameters of Grounded Correspondence for main results on MOVi-D, MOVi-E, and YouTube-VIS 2021 datasets.

| Hyperparameter | MOVi-D | MOVi-E | YouTube-VIS |
|---|---|---|---|
| **Training Configuration** | | | |
| Training Steps | 100k | 100k | 100k |
| Batch Size | 8 | 8 | 8 |
| Training Segment Length | 4 | 4 | 4 |
| Learning Rate Warmup Steps | 2500 | 2500 | 2500 |
| Optimizer | Adam | Adam | Adam |
| Peak Learning Rate | 0.0004 | 0.0004 | 0.0008 |
| Exponential Decay | 100k | 100k | 100k |
| Gradient Norm Clipping | 0.05 | 0.05 | 0.05 |
| **Vision Backbone** | | | |
| ViT Architecture | DINOv2 Base | DINOv2 Base | DINOv2 Base |
| Patch Size | 14 | 14 | 14 |
| Feature Dimension ($D_{\text{feat}}$) | 768 | 768 | 768 |
| Frozen Backbone | True | True | True |
| **Image Specifications** | | | |
| Image / Crop Size | 336 | 336 | 518 |
| Cropping Strategy | Full | Full | Rand. Center Crop |
| Image Tokens | 576 | 576 | 1369 |
| **Grounded Saliency Initialization** | | | |
| Saliency Metric | Grounded Saliency Metric | Grounded Saliency Metric | LGrounded Saliency Metric |
| Background Penalty ($\alpha$) | 0.5 | 1.0 | 0.5 |
| Spatial Radius ($r$) | 1 | 1 | 2 |
| **Slot Attention** | | | |
| Slots | 15 | 15 | 7 |
| Iterations (first / other frames) | 3 / 2 | 3 / 2 | 3 / 2 |
| Slot Dimension ($D_{\text{slots}}$) | 128 | 128 | 64 |
| **Temporal Correspondence** | | | |
| Method | Hungarian | Hungarian | Hungarian |
| Similarity Metric | Cosine | Cosine | Cosine |
| Learnable Parameters | 0 | 0 | 0 |
| **Decoder** | | | |
| Type | MLP | MLP | MLP |
| **Loss Parameters** | | | |
| Softmax Temperature ($\tau$) | 0.1 | 0.1 | 0.1 |
| Slot-Slot Contrast Weight ($\alpha$) | 0.5 | 0.5 | 0.5 |
| Feature Recon. Weight | 1.0 | 1.0 | 1.0 |

