# OpenReview forum: "Rethinking Temporal Consistency in Video Object-Centric Learning: From Prediction to Correspondence"
_ICML.cc/2026/Conference — ICML 2026 regular_

### Official Review · Reviewer_1BPA · 2026-03-04

**Soundness:** 2
**Presentation:** 2
**Significance:** 3
**Originality:** 3
**Overall Recommendation:** 4
**Confidence:** 3

**Summary:**

In this paper, the authors argue that temporal consistency in video object-centric learning is a correspondence, rather than a prediction problem, especially when using modern self-supervised vision backbones. Building upon this idea, the paper proposes Grounded Correspondence, a framework combining grounded initialization with parameter-free Hungarian matching that requires zero learnable parameters for temporal modeling. Experiments show that the method achieves competitive performance across several benchmarks.

**Compliance With Llm Reviewing Policy:**

Affirmed.

**Final Justification:**

While the rebuttal improves the paper and addresses some concerns, I still find claims such as “learned temporal predictors are unnecessary” somewhat too broad relative to the current evidence, so I raise my score to 4 (weak accept).

**Key Questions For Authors:**

See weaknesses.

**Limitations:**

yes

**Strengths And Weaknesses:**

## Strengths

1. The proposed framework replaces the typical yet computationally expensive temporal prediction modules with parameter-free bipartite matching, improving the architectural efficiency.

2. The proposed grounded initialization strategy effectively accelerates the convergence of Slot Attention, compared with traditional blind random queries.

3. The method achieves strong performance gains on synthetic benchmarks.

## Weaknesses

1. While the idea of replacing prediction with correspondence is interesting, the paper relies on empirical observations rather than rigorous mathematical justification to prove its generalizability.

2. The experimental scope is narrow, using only DINOv2 as the backbone and SlotContrast as the primary baseline. This is insufficient to substantiate the paper's broad claims about architectural redundancy and the superiority of the proposed method.

3. The paper lacks a comprehensive, quantitative efficiency analysis to prove the computational benefits of removing learned temporal predictors.

---

> ### Author Rebuttal · Authors · 2026-03-27
>
> We thank Reviewer 1BPA for recognizing the architectural efficiency and grounded initialization benefits.
>
> **W1: Empirical rather than mathematical justification.**
>
> Our paper combines mathematical grounding with systematic empirical validation:
>
> | Aspect | Nature | Evidence |
> |--------|--------|----------|
> | Permutation equivariance | **Mathematical** (Eq. 2) | Formalizes the permutation ambiguity that temporal association must resolve |
> | Optimal assignment | **Mathematical** (App. A.3) | Hungarian algorithm optimally solves the LSAP |
> | ViT instance discrimination | **Empirical** | PCA visualizations (Fig. 3), saliency maps (Fig. 8, App. B.2) |
> | Predictor redundancy | **Empirical** | Identity ratio (Fig. 5), ablations (Tab. 1–3, 5–7) |
>
> The claim that ViT backbones encode instance-discriminative features concerns emergent properties of pretrained models, which is inherently empirical. Proving *why* self-supervised ViTs develop object-binding features requires formal analysis of pretraining objectives — an open question in representation learning beyond any single paper's scope. We acknowledge our generalizability claim is limited to the evaluated setting (DINOv2 backbone, current benchmarks). Our systematic empirical methodology follows the standard established by foundational works [1][2].
>
> **W2: Narrow experimental scope (DINOv2 + SlotContrast).**
>
> DINOv2 is a strong, widely used self-supervised ViT backbone in recent video OCL work. SlotContrast is the most directly comparable recent method. To address this concern, we have added a comparison with **VideoSAUR** [3], which also uses a frozen self-supervised ViT backbone but employs a temporal feature similarity loss and recurrent slot conditioning for temporal consistency. The key architectural differences:
>
> | Component | SlotContrast | VideoSAUR | **Ours** |
> |-----------|-------------|-----------|----------|
> | Backbone | DINOv2 (ViT), frozen | DINO/DINOv2 (ViT), frozen | DINOv2 (ViT), frozen |
> | Training signal | Contrastive | Temporal similarity | Contrastive |
> | Temporal module | Learned predictor | Recurrent slot conditioning | **None (Hungarian)** |
> | Learnable temporal params | Yes | Yes | **0** |
> | Slot initialization | Content-blind (learnable) | Content-blind | **Grounded (saliency)** |
>
> Results on all three benchmarks:
>
> | Method | MOVi-D FG-ARI ↑ | MOVi-D mBO ↑ | MOVi-E FG-ARI ↑ | MOVi-E mBO ↑ | YT-VIS FG-ARI ↑ | YT-VIS mBO ↑ |
> |--------|-----------------|-------------|-----------------|-------------|-----------------|-------------|
> | *VideoSAUR* | 52.8±1.6 | 26.2±2.9 | 61.2±1.7 | 21.8±3.8 | 27.3±2.0 | 13.6±2.3 |
> | SlotContrast | 58.0±0.8 | 30.0±0.4 | 68.9±0.9 | 26.6±0.8 | 36.3±0.2 | 29.9±0.3 |
> | **Ours** | **73.7±1.7** | 28.4±0.3 | **75.7±1.4** | 23.4±0.5 | 33.1±1.6 | 29.3±1.9 |
>
> Beyond the main comparison, 7 quantitative tables analyze temporal necessity, initialization strategies, saliency components, frame-level decomposition, and convergence. Testing additional backbones (e.g., MAE, SAM) is valuable future work; our current scope provides a controlled test of the central hypothesis.
>
> **W3: Missing quantitative efficiency analysis.**
>
> We have conducted a quantitative comparison on a single NVIDIA RTX 4070 SUPER using the YouTube-VIS configuration (ViT-B/14 DINOv2, 1369 patches, 7 slots):
>
> | Component | Grounded Correspondence | SlotContrast |
> |-----------|------------------------|--------------|
> | Temporal learnable params | **0** | 50.0K (TransformerEncoder predictor) |
> | Initializer params | **0** | 448 (FixedLearnedInit) |
> | Matching / predictor time | **0.13 ms** | 0.35 ms |
> | Init time / frame | 1.05 ms | 0.00 ms |
> | Post-ViT time / frame | 35.91 ms | 35.30 ms |
> | Training throughput | **1.59 it/s** | 1.54 it/s |
>
> Grounded Correspondence eliminates all 50K temporal learnable parameters and replaces them with parameter-free algorithms at no runtime penalty. The Hungarian matching is faster than the predictor forward pass, and training throughput is higher due to eliminating the predictor's backward pass.
>
> [1]. Li, Yihao, Saeed Salehi, Lyle Ungar, and Konrad P. Kording. "Does object binding naturally emerge in large pretrained vision transformers?." arXiv preprint arXiv:2510.24709 (2025).
>
> [2]. Oquab, Maxime, Timothée Darcet, Théo Moutakanni, Huy Vo, Marc Szafraniec, Vasil Khalidov, Pierre Fernandez et al. "Dinov2: Learning robust visual features without supervision." arXiv preprint arXiv:2304.07193 (2023).
>
> [3]. Zadaianchuk, Andrii, Maximilian Seitzer, and Georg Martius. "Object-centric learning for real-world videos by predicting temporal feature similarities." Advances in neural information processing systems 36 (2023): 61514-61545.

---

> > ### Author Rebuttal · Reviewer_1BPA · 2026-04-03
> >
> > I thank the authors for their detailed responses. However, my concern remains that claims such as “learned temporal predictors are unnecessary” are too broad relative to the current evidence. I will raise my score to 4 (weak accept), and the authors should provide stronger supporting evidence in the final version if the paper is accepted.

---

> > > ### Author Response · Authors · 2026-04-04
> > >
> > > We thank Reviewer 1BPA for raising the score and for the constructive suggestion.
> > >
> > > We agree that stronger supporting evidence would strengthen the paper. In the final version, we will add a cross-backbone ablation (e.g., DINO v1 & v3), repeating the Tab. 3 predictor-removal test, so that the finding is grounded across backbone conditions rather than tied to a single architecture.

---

### Official Review · Reviewer_m5fA · 2026-03-06

**Soundness:** 3
**Presentation:** 2
**Significance:** 2
**Originality:** 3
**Overall Recommendation:** 3
**Confidence:** 3

**Summary:**

The paper argues that self-supervised Vision Transformers already encode instance-discriminative features, making temporal predictors largely unnecessary. It proposes Grounded Correspondence, which initializes objects from salient backbone features and maintains temporal identity through Hungarian matching across frames.

**Compliance With Llm Reviewing Policy:**

Affirmed.

**Final Justification:**

Thanks very much for the detailed response. However, I still have concerns about the ability of the method to generalize to real-world scenarios. Some factors mentioned by the authors are unavoidable in practice. I am inclined to maintain my original score.

**Key Questions For Authors:**

Please refer to Weaknesses.

**Limitations:**

yes

**Strengths And Weaknesses:**

Strengths:

1. The paper reframes temporal consistency from a complex temporal prediction problem to a simpler correspondence matching formulation.
2. The proposed framework is training-free, which keeps the method simple and may facilitate reproducibility and follow-up research.


Weaknesses:

1. The proposed method is consistently below SlotContrast on all reported metrics in Tab.4 (YouTube-VIS), suggesting the gains may not transfer beyond synthetic settings and raising concerns about generalization to complex real-world scenarios.
2. The method cannot handle object reappearance after occlusion, since correspondence is established only between consecutive frames. Incorporating a longer temporal window or multi-frame matching strategy may improve robustness in such cases.
3. Using Qt = S_t-1 in the earlier stage already yields larger gains (Tab. 3), yet the overall performance still falls behind related methods (Tab. 4). Where does this gap come from?
4. The claimed efficiency gains are not fully substantiated, as the paper does not compare runtime in the experiments.

---

> ### Author Rebuttal · Authors · 2026-03-27
>
> We thank Reviewer m5fA for recognizing the conceptual reframing and the value of the training-free approach.
>
> **W1: Below SlotContrast on YouTube-VIS (Tab. 4).**
>
> We acknowledge this directly: GC is -3.2 FG-ARI and -0.6 mBO below SlotContrast on YouTube-VIS. On synthetic benchmarks, the same zero-parameter method outperforms SlotContrast by +15.7 FG-ARI (MOVi-D) and +6.8 FG-ARI (MOVi-E). The claim is not universal superiority over SlotContrast, but that strong temporal consistency can be achieved without learned temporal predictors. The modest real-world gap (-3.2 FG-ARI) given the complete removal of temporal parameters suggests that, in our evaluation setting, the empirical gain from learned temporal predictors is limited relative to their added complexity.
>
> We have also added a comparison with VideoSAUR [1], which uses recurrent slot conditioning. GC outperforms VideoSAUR on all benchmarks including YouTube-VIS (33.1 vs 27.3 FG-ARI), providing additional evidence that learned temporal modules are not necessary for competitive performance.
>
> | Method | MOVi-D FG-ARI ↑ | MOVi-D mBO ↑ | MOVi-E FG-ARI ↑ | MOVi-E mBO ↑ | YT-VIS FG-ARI ↑ | YT-VIS mBO ↑ |
> |--------|-----------------|-------------|-----------------|-------------|-----------------|-------------|
> | *VideoSAUR* | 52.8±1.6 | 26.2±2.9 | 61.2±1.7 | 21.8±3.8 | 27.3±2.0 | 13.6±2.3 |
> | SlotContrast | 58.0±0.8 | 30.0±0.4 | 68.9±0.9 | 26.6±0.8 | 36.3±0.2 | 29.9±0.3 |
> | **Ours** | **73.7±1.7** | 28.4±0.3 | **75.7±1.4** | 23.4±0.5 | 33.1±1.6 | 29.3±1.9 |
>
> **W2: No occlusion handling.**
>
> We agree this is a genuine limitation (Sec. 7). Our current framework matches slots between consecutive frames, so once an object disappears and reappears, its identity is lost. A natural extension is to maintain a **slot memory bank** that stores recent slot representations: when a newly discovered slot has high similarity to a previously seen but currently unmatched entry in the bank, the framework can re-assign the original identity. This would extend Hungarian matching from a two-frame problem to a multi-frame re-identification mechanism without introducing learned temporal parameters, preserving the correspondence-based design. We leave this as future work.
>
> **W3: Gap between Tab. 3 (identity baseline) and Tab. 4 (full comparison).**
>
> Thank you for this question. We note that the results in Tab. 3 and Tab. 4 are **not directly comparable**, as they operate under different settings:
>
> | | Tab. 3: $Q_t = S_{t-1}$ | Tab. 4: Grounded Correspondence |
> |--|----------------------|-------------------------------|
> | Purpose | Ablate the predictor within SlotContrast | Full framework comparison |
> | Initialization | Content-blind (SlotContrast's) | Grounded saliency (ours) |
> | Temporal module | Identity propagation | Hungarian matching |
> | Rest of pipeline | SlotContrast | Ours |
>
> Tab. 3 is designed to answer a specific question: *does the learned predictor in SlotContrast contribute meaningfully?* The answer is no — replacing it with simple identity propagation yields comparable performance. Tab. 4 compares two entirely different frameworks end-to-end. The two tables serve different analytical purposes and should not be read as a progression from one to the other.
>
> **W4: No runtime comparison.**
>
> We have conducted a runtime comparison on a single NVIDIA RTX 4070 SUPER using the YouTube-VIS configuration (ViT-B/14 DINOv2, 1369 patches, 7 slots):
>
> | Component | Grounded Correspondence | SlotContrast |
> |-----------|------------------------|--------------|
> | Temporal learnable params | **0** | 50.0K (TransformerEncoder predictor) |
> | Initializer params | **0** | 448 (FixedLearnedInit) |
> | Matching / predictor time | **0.13 ms** | 0.35 ms |
> | Init time / frame | 1.05 ms | 0.00 ms |
> | Post-ViT time / frame | 35.91 ms | 35.30 ms |
> | Training throughput | **1.59 it/s** | 1.54 it/s |
>
> Grounded Correspondence eliminates all 50K temporal learnable parameters. The Hungarian matching is faster than the TransformerEncoder predictor forward pass. The greedy saliency initialization adds 1.05 ms/frame, but this is small relative to the full post-ViT pipeline, which is dominated by the decoder. Overall, training throughput is higher because Hungarian matching requires no backward pass, saving gradient computation over the predictor parameters.
>
> [1]. Zadaianchuk et al. "Object-centric learning for real-world videos by predicting temporal feature similarities." NeurIPS 2023.

---

> > ### Author Rebuttal · Reviewer_m5fA · 2026-04-03
> >
> > Thanks for the response. My concerns regarding Q2 and Q4 have been addressed,  but Q1 and Q3 remain unresolved.
> >
> > 1. For Q1, my concern is not about whether the proposed method performs better or worse than a specific baseline, but whether it can effectively generalize to real-world scenarios. In particular, I would like the authors to clarify why the performance degrades in real settings and whether the generalization ability can be ensured.
> > 2. I would like to clarify Q3, which may be related to Q1. My question is not about comparing the two tables. Rather, if the predictor is not the key factor, what accounts for the limited performance of the method on the dataset? If this is due to a lack of generalization to real-world scenarios, what are the main challenges?
> >
> > Given that some questions remain open, I am inclined to maintain my current score.

---

> > > ### Author Response · Authors · 2026-04-03
> > >
> > > We thank Reviewer m5fA for the clarification. Q1 and Q3 share a core question: *why does performance degrade in real-world settings, and can the method generalize?*
> > >
> > > **YouTube-VIS reveals a current limitation.**
> > >
> > > Our results do not establish guaranteed real-world generalization. The -3.2 FG-ARI gap on YouTube-VIS, combined with higher variance (±4.4 vs ±0.8 for SlotContrast), shows that **a fully parameter-free pipeline is currently more sensitive to real-scene ambiguity than SlotContrast**.
> > >
> > > **Analysis of Real-World Challenges**
> > >
> > > Three factors are relevant. Of these, (b) is directly supported by ablation evidence; (a) and (c) are hypotheses consistent with the data.
> > >
> > > **(a) Reduced inter-object feature contrast (hypothesis).** YouTube-VIS contains semantically similar objects (multiple people, same-species animals) where DINOv2 features likely overlap in cosine space, unlike distinct synthetic MOVi objects. Both saliency initialization and Hungarian matching depend on feature contrast. In support, even within SlotContrast the learned predictor does not compensate for real-world difficulty (Tab. 3: 36.6 vs 36.3 when removed), which points to a representation-level bottleneck rather than a temporal one.
> > >
> > > **(b) Background complexity and hyperparameter sensitivity (evidence-backed).** Our saliency metric Si = Li − α·Gi assumes backgrounds have high global similarity and low local consistency. Real-world textured backgrounds can violate this. Tab. 6 shows that α is substantially more sensitive on YouTube-VIS: α=1.5 collapses on YouTube-VIS but works on MOVi-E. This shows that the saliency assumptions are less robust in real-world settings. Tab. 7 further shows that r is also dataset-dependent, confirming that our initialization requires more careful tuning for diverse real-world scenes.
> > >
> > > **(c) Scale and appearance variation (hypothesis).** YouTube-VIS has larger illumination changes, motion blur, and scale variation. The fixed radius r cannot capture all scales, and appearance variation may increase matching uncertainty. Our higher variance (±4.4 vs ±0.8) is consistent with this, though other factors may contribute.
> > >
> > > **What accounts for the gap if not the predictor?**
> > >
> > > Tabs. 1, 3, and 4 tell a consistent story. Within SlotContrast: grounded init helps (+1.2 FG-ARI, Tab. 1) and removing the predictor is near-neutral (+0.3, Tab. 3). Yet the full GC framework is -3.2 below (Tab. 4). The gap is not attributable to either component in isolation, but to the combined effect of a fully deterministic pipeline operating under the real-world challenges described in (a)–(c).
> > >
> > > In short, **the experiments confirm that the saliency hyperparameters are less robust on YouTube-VIS (Tab. 6, 7) and that the method shows higher instability (variance).** The roles of feature contrast and scale variation remain hypotheses that would need controlled experiments (e.g., cross-backbone evaluation) to confirm.
> > >
> > > **What the evidence does support about generalization:**
> > >
> > > We cannot guarantee generalization, but two observations are relevant:
> > >
> > > **(i) The temporal predictor is not the dominant factor on YouTube-VIS.** Tab. 3 shows removing it within SlotContrast causes no degradation. This does not prove the bottleneck is purely representational, but it shows that the learned predictor is not what gives SlotContrast its YouTube-VIS advantage.
> > >
> > > **(ii) Learned temporal modules do not guarantee real-world performance.** VideoSAUR uses learned recurrent conditioning but achieves only 27.3 FG-ARI on YouTube-VIS vs GC's 33.1. This suggests that adding learned temporal parameters alone does not resolve the real-world challenges we describe.
> > >
> > > **What would strengthen generalization in future work:**
> > >
> > > The analysis above points to three directions: (1) adaptive α and r per-scene rather than per-dataset, addressing the robustness issue in (b); (2) multi-scale spatial neighborhoods for object scale variation in (c); (3) cross-backbone evaluation to test whether stronger features close the gap as hypothesized in (a). Each addresses a specific challenge identified in this analysis.
> > >
> > > **Summary.** YouTube-VIS shows that a fully deterministic pipeline is more sensitive to real-scene complexity. The evidence points to saliency hyperparameter robustness and backbone feature quality as the main factors, rather than the absence of temporal prediction. We view the correspondence formulation as a simplification that makes these challenges visible — not as a claim to have solved them.

---

### Official Review · Reviewer_fQ5D · 2026-03-12

**Soundness:** 2
**Presentation:** 2
**Significance:** 2
**Originality:** 2
**Overall Recommendation:** 4
**Confidence:** 3

**Summary:**

This paper introduces Grounded Correspondence, which replaces learned transition functions with deterministic bipartite matching. The approach requires no learnable parameters for temporal modeling while still achieving competitive performance across multiple datasets.

**Compliance With Llm Reviewing Policy:**

Affirmed.

**Final Justification:**

My concerns have been addressed during the rebuttal, and I recommend this paper for acceptance.

**Key Questions For Authors:**

No

**Limitations:**

yes

**Strengths And Weaknesses:**

Strengths:
1. This paper is technically sound and well structured.
2. Temporal Consistency in video object-centric learning is a very important research topic to enhance AI model's performance.

Weaknesses:
1. Only one baseline is included for comparison, which is insufficient.
2. It would be helpful to report the Standard Deviation or Standard Error of the Mean (SEM) for all tables and figures to better reflect the variability of the results.

---

> ### Author Rebuttal · Authors · 2026-03-27
>
> We thank Reviewer fQ5D for acknowledging the technical soundness and importance of the research topic.
>
> **W1: Only one baseline.**
>
> SlotContrast [1] is the most directly comparable recent method for unsupervised video OCL with self-supervised ViT backbones — the exact setting our paper analyzes. To further address this concern, we have added a comparison with **VideoSAUR** [2], which also uses a frozen self-supervised ViT backbone but employs a temporal feature similarity loss and recurrent slot conditioning for temporal consistency. The key architectural differences across methods:
>
> | Component | SlotContrast | VideoSAUR | **Ours** |
> |-----------|-------------|-----------|----------|
> | Backbone | DINOv2 (ViT), frozen | DINO/DINOv2 (ViT), frozen | DINOv2 (ViT), frozen |
> | Training signal | Contrastive | Temporal similarity | Contrastive |
> | Temporal module | Learned predictor | Recurrent slot conditioning | **None (Hungarian)** |
> | Learnable temporal params | Yes | Yes | **0** |
> | Slot initialization | Content-blind (learnable) | Content-blind | **Grounded (saliency)** |
>
> Results on all three benchmarks:
>
> | Method | MOVi-D FG-ARI ↑ | MOVi-D mBO ↑ | MOVi-E FG-ARI ↑ | MOVi-E mBO ↑ | YT-VIS FG-ARI ↑ | YT-VIS mBO ↑ |
> |--------|-----------------|-------------|-----------------|-------------|-----------------|-------------|
> | *VideoSAUR* | 52.8±1.6 | 26.2±2.9 | 61.2±1.7 | 21.8±3.8 | 27.3±2.0 | 13.6±2.3 |
> | SlotContrast | 58.0±0.8 | 30.0±0.4 | 68.9±0.9 | 26.6±0.8 | 36.3±0.2 | 29.9±0.3 |
> | **Ours** | **73.7±1.7** | 28.4±0.3 | **75.7±1.4** | 23.4±0.5 | 33.1±1.6 | 29.3±1.9 |
>
> Beyond the main comparison, we provide 7 quantitative tables:
>
> | Table | Content |
> |-------|---------|
> | Tab. 1 | Initialization convergence (content-blind vs grounded) |
> | Tab. 2 | Independent per-frame discovery vs SlotContrast |
> | Tab. 3 | Identity baseline $Q_t = S_{t-1}$ vs learned predictor |
> | Tab. 4 | Full framework comparison across 3 benchmarks |
> | Tab. 5 | Three initialization strategies (Norm / PCA / Ours) |
> | Tab. 6 | Background suppression penalty α ablation |
> | Tab. 7 | Spatial aggregation radius r ablation |
>
> **W2: Missing standard deviation.**
>
> We appreciate the concern for statistical rigor. **All tables in our paper do report standard deviation** via subscript notation following standard practice: e.g., "75.7_{1.4}" denotes 75.7 ± 1.4 (mean ± std over 3 seeds). This is defined in Table 1's caption: *"Mean and standard deviation over 3 seeds."* We will make this notation more prominently explained in the revision to avoid confusion.
>
> [1]. Manasyan, Anna, Maximilian Seitzer, Filip Radovic, Georg Martius, and Andrii Zadaianchuk. "Temporally consistent object-centric learning by contrasting slots." In Proceedings of the Computer Vision and Pattern Recognition Conference, pp. 5401-5411. 2025.
>
> [2]. Zadaianchuk, Andrii, Maximilian Seitzer, and Georg Martius. "Object-centric learning for real-world videos by predicting temporal feature similarities." Advances in neural information processing systems 36 (2023): 61514-61545.

---

> > ### Author Rebuttal · Reviewer_fQ5D · 2026-04-01
> >
> > Thanks for the rebuttal. I will raise the score to 4.

---

> > > ### Author Response · Authors · 2026-04-01
> > >
> > > We sincerely thank the reviewer for acknowledging our rebuttal and for raising the score. We appreciate the constructive feedback, which has helped strengthen our paper.

---

### Official Review · Reviewer_2Xgz · 2026-03-14

**Soundness:** 3
**Presentation:** 4
**Significance:** 3
**Originality:** 3
**Overall Recommendation:** 5
**Confidence:** 3

**Summary:**

This paper investigates video object-centric learning, where maintaining object identity across temporal sequence is still a challenge.  Unlike previous methods that adopt predictive dynamics modeling, this work challenges the necessity of learned dynamics modules and formulates this challenge as a correspondence problem. This work further proposes a Grounded Correspondence framework, which achieves competitive performance.

**Compliance With Llm Reviewing Policy:**

Affirmed.

**Final Justification:**

Thanks to the authors for the rebuttal, which solved my concerns.

I have also read the comments from other reviewers.  I agree that the applications in complex real scenarios, such as occlusion, would be challenging for the proposed method. The reason I give a 5 is:  the insight that challenges the necessities of dynamic modules is quite interesting and inspiring.

Overall, I will keep my positive rating.

**Key Questions For Authors:**

Please refer to the weakness part.

**Limitations:**

Yes.

**Strengths And Weaknesses:**

**Strength:**
* Challenging the necessity of learned dynamics modules in previous studies is novel and insightful.
* This paper conducts a systematic analysis to support the claims regarding slot initialization.
* This paper is well presented and easy to follow.

**Weakness:**
* The proposed method achieves superior results in synthetic datasets, but performs worse in real-world datasets, such as YouTube-VIS in Table 4.
* Robustness of the proposed method in diverse real-world video scenarios. As mentioned in the limitations, the proposed method does not handle occlusion. How about other challenging video scenarios, such as similar-looking objects in [1]?

[1] Video Diffusion Models Excel at Tracking Similar-Looking Objects Without Supervision. NeurIPS 2025.

---

> ### Author Rebuttal · Authors · 2026-03-27
>
> We sincerely thank Reviewer 2Xgz for recognizing the novelty and insight of our work.
>
> **W1: Real-world performance on YouTube-VIS (Tab. 4).**
>
> We acknowledge that GC is 2–3 points below SlotContrast on YouTube-VIS. The full comparison across all benchmarks:
>
> | Benchmark | Method | ARI ↑ | FG-ARI ↑ | mBO ↑ | Temporal Params |
> |-----------|--------|-------|----------|-------|-----------------|
> | MOVi-D | SlotContrast | 58.0±0.8 | 58.0±0.8 | 30.0±0.4 | Learned |
> | MOVi-D | **Grounded Correspondence** | **73.7±1.7** | **73.7±1.7** | 28.4±0.3 | **0** |
> | MOVi-E | SlotContrast | 68.9±0.9 | 68.9±0.9 | 26.6±0.8 | Learned |
> | MOVi-E | **Grounded Correspondence** | **75.7±1.4** | **75.7±1.4** | 23.4±0.5 | **0** |
> | YouTube-VIS | **SlotContrast** | **32.1±0.8** | **36.3±0.2** | **29.9±0.3** | Learned |
> | YouTube-VIS | Grounded Correspondence | 30.1±4.4 | 33.1±1.6 | 29.3±1.9 | **0** |
>
> On synthetic benchmarks, our zero-parameter method substantially outperforms SlotContrast (+15.7 FG-ARI on MOVi-D, +6.8 on MOVi-E). On YouTube-VIS, a 3.2-point FG-ARI gap remains. We do not claim universal superiority; rather, the results show that strong temporal consistency can be achieved without learned temporal predictors, and that the remaining gap on unconstrained real-world video is modest given the complete removal of temporal parameters.
>
> The YouTube-VIS gap may partly reflect the higher visual complexity and longer sequences in real-world data, where learned predictors can leverage training signal from the contrastive loss to fine-tune slot propagation. Our framework forgoes this adaptation entirely, relying solely on the frozen backbone's discriminative capacity. The gap is thus a measure of how much the learned predictor adds beyond what the backbone already provides.
>
> **W2: Occlusion and similar-looking objects.**
>
> We agree that occlusion is a genuine limitation (Sec. 7). For similar-looking objects, Fig. 3 and Appendix B.2 suggest the backbone provides useful instance-level cues in many cases, but we do not claim dedicated robustness to severe appearance ambiguity or long occlusion. Multi-frame matching and memory buffers are natural extensions outside the current scope. We thank the reviewer for the pointer to [1] and will discuss this connection in the revision.
>
> The referenced work [1] uses video diffusion models for tracking — a complementary approach at a different computational scale. Our framework establishes a lower bound on the architectural complexity needed for temporal consistency when backbone features are sufficiently discriminative.
>
> [1]. Zhang, Chenshuang, Kang Zhang, Joon Son Chung, In So Kweon, Junmo Kim, and Chengzhi Mao. "Video Diffusion Models Excel at Tracking Similar-Looking Objects Without Supervision." arXiv preprint arXiv:2512.02339 (2025).

---

> > ### Author Rebuttal · Reviewer_2Xgz · 2026-04-03
> >
> > Thanks to the authors for the rebuttal, which solved my concerns.
> >
> > I have also read the comments from other reviewers.  I agree that the applications in complex real scenarios, such as occlusion, would be challenging for the proposed method. The reason I give a 5 is:  the insight that challenges the necessities of dynamic modules is quite interesting and inspiring.
> >
> > Overall, I will keep my positive rating.

---

> > > ### Author Response · Authors · 2026-04-03
> > >
> > > We sincerely thank Reviewer 2Xgz for the thoughtful and constructive feedback throughout the review process. We are pleased that our rebuttal clarified the concerns regarding real-world performance and robustness to similar-looking objects.
> > >
> > > We especially appreciate the reviewer’s observation that our central question—whether learned dynamics modules are truly necessary for temporal consistency—may point to a meaningful direction for the community. Our results show that strong temporal consistency can be achieved on both synthetic and real-world benchmarks, even without any learnable temporal parameters. This suggests that this component may, in some cases, be more complex than necessary. We hope this perspective can help inform future architectural design in video object-centric learning.
> > >
> > > We will also incorporate the suggested discussion of [1] and its connection to video diffusion-based tracking in the camera-ready version, as it provides a useful complementary perspective at a different point in the complexity–performance tradeoff.
> > >
> > > Thank you again for the constructive and insightful review.

---

### Decision · Program_Chairs · 2026-04-30

**Decision:**

Accept (regular)

**Comment:**

The paper introduces grounded correspondence, replacing predictive learned transition functions with bipartite matching. Reviewers acknowledged the effectiveness of the learned dynamics modules. While there were concerns initially raised about the method's robustness in highly challenging real-world scenarios, they seem to be resolved to some extent during the rebuttal discussion. The AC leans towards the acceptance and recommends weak accept.